

# *EquiRate*: balanced rating injection approach for popularity bias mitigation in recommender systems

Mert Gulsoy[1,2], Emre Yalcin[3] and Alper Bilge[2]

[1] Distance Education Research Center, Alaaddin Keykubat University, Antalya, Turkey
[2] Computer Engineering Department, Akdeniz University, Antalya, Turkey
[3] Computer Engineering Department, Sivas Cumhuriyet University, Sivas, Turkey

## ABSTRACT

Recommender systems often suffer from popularity bias problem, favoring popular items and overshadowing less known or niche content, which limits recommendation diversity and content exposure. The root reason for this issue is the imbalances in the rating distribution; a few popular items receive a disproportionately large share of interactions, while the vast majority garner relatively few. In this study, we propose the *EquiRate* method as a pre-processing approach, addressing this problem by injecting synthetic ratings into less popular items to make the dataset regarding rating distribution more balanced. More specifically, this method utilizes several synthetic rating injection and synthetic rating generation strategies: (*i*) the first ones focus on determining which items to inject synthetic ratings into and calculating the total number of these ratings, while (*ii*) the second ones concentrate on computing the concrete values of the ratings to be included. We also introduce a holistic and highly efficient evaluation metric, *i.e.*, the *FusionIndex*, concurrently measuring accuracy and several beyond-accuracy aspects of recommendations. The experiments realized on three benchmark datasets conclude that several *EquiRate*'s variants, with proper parameter-tuning, effectively reduce popularity bias and enhance recommendation diversity. We also observe that some prominent popularity-debiasing methods, when assessed using the *FusionIndex*, often fail to balance the referrals' accuracy and beyond-accuracy factors. On the other hand, our best-performing *EquiRate* variants significantly outperform the existing methods regarding the *FusionIndex*, and their superiority is more apparent for the high-dimension data collections, which are more realistic for real-world scenarios.

# INTRODUCTION

Recommender systems have significantly reshaped how users interact with digital content and services by providing personalized recommendations (*Bobadilla et al., 2013*; *Zhang, Lu & Jin, 2021*). These systems, at their core, are designed to predict and suggest items, such as movies, books, or products, that are likely to be of interest to the user, based on various data inputs (*Wang et al., 2023*). Their application spans across numerous domains,

Corresponding author
Emre Yalcin,
eyalcin@cumhuriyet.edu.tr

including e-commerce, entertainment, and social networking, fundamentally enhancing user experience and engagement (*Koren, 2009*; *Li, Chen & Raghunathan, 2018*). By analyzing large datasets, including user preferences, behavior, and item characteristics, recommender systems personalize content delivery, making them indispensable in today's information-rich digital landscape. This technology not only drives user satisfaction but also boosts business metrics by facilitating better-targeted content and advertising. The burgeoning field of recommender systems is continuously evolving, employing sophisticated algorithms ranging from collaborative filtering to deep learning, showcasing the relentless pursuit of more accurate, context-aware, and user-centric recommendation strategies.

Despite their widespread adoption and success, these systems face several significant challenges that can impede their effectiveness (*Ricci, Rokach & Shapira, 2015*). One of the primary issues is data sparsity; even with large datasets, the user-item interactions are often too few, leading to challenges in generating accurate recommendations. Another critical hurdle is the cold-start problem, where new users or items in the system lack sufficient interaction data to make reliable recommendations. Robustness is also a concern, as these systems must be resilient to manipulations or anomalies in the data to maintain the integrity of their suggestions. Furthermore, general bias issues, such as user or item biases, can skew the recommendations towards certain products or users, thereby limiting the diversity of recommendations (*Chen et al., 2023*).

Among these challenges, the issue of popularity bias is particularly detrimental (*Abdollahpouri, Burke & Mobasher, 2019*; *Yalcin & Bilge, 2021*; *Elahi et al., 2021*). This bias causes recommender systems to disproportionately favor popular items, resulting in a narrow concentration of recommendations. Such a trend not only undermines the visibility of less popular or new items but also stifles diversity and novelty in the recommendations. The negative impact of popularity bias extends beyond just reduced item exposure can lead to a homogenization of user experience, where the rich and varied tastes of users are not adequately catered to. This homogeneity can also perpetuate a feedback loop, where popular items become even more dominant, further marginalizing niche content. The result is a diminished overall user experience, with limited opportunities for content discovery and personalization. Addressing this popularity bias is, therefore, a key objective in the advancement of recommender systems, aiming to create a more balanced, diverse, and genuinely user-centric recommendation landscape (*Abdollahpouri et al., 2021*; *Boratto, Fenu & Marras, 2021*; *Yalcin & Bilge, 2022*).

The root cause of the popularity bias problem in recommender systems can be traced back to the imbalanced distribution of user ratings. Typically, a small subset of items receives a disproportionately large share of interactions and ratings from users, while the vast majority of items garner relatively few. This imbalance is often reflective of real-world user behavior, where popular items naturally attract more attention and feedback. However, when such skewed data is fed into recommender systems, it leads to an amplification of the popularity effect. The algorithms, aiming to maximize accuracy and relevance, tend to favor items with a higher volume of interactions, assuming they are more likely to be relevant to a broader user base. Figure 1 illustrates the imbalances in

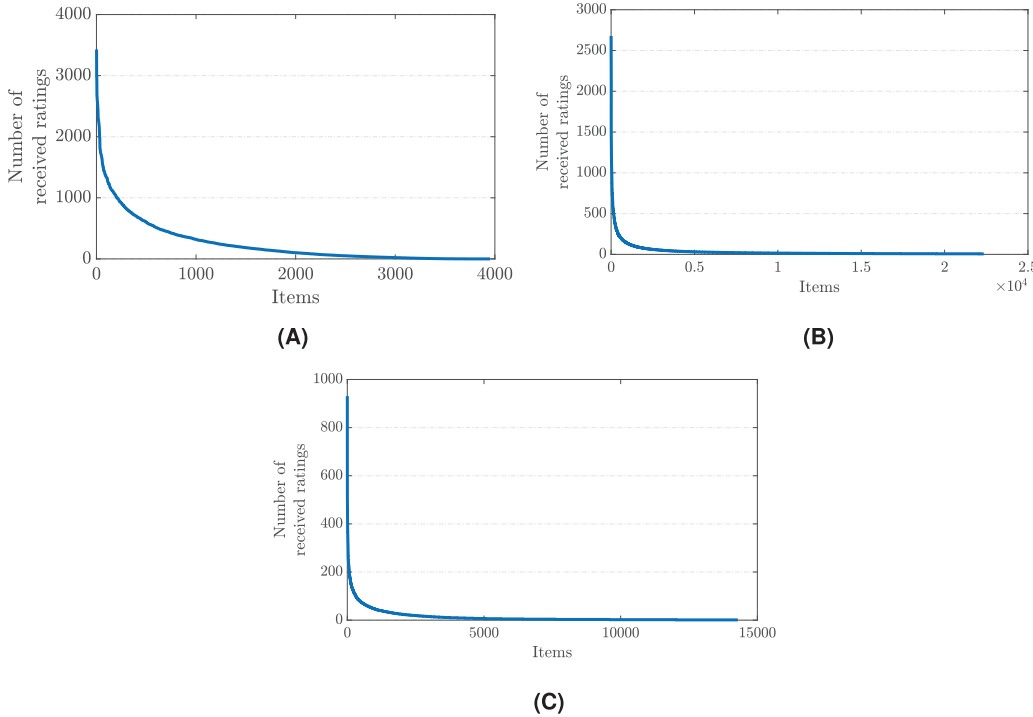

**Figure 1** The rating distribution across items in (A) the MovieLens-1M (movie reviews), (B) Douban Book (book reviews), and (C) Yelp (business reviews) datasets (*Gulsoy, Yalcin & Bilge, 2023*). Details of these datasets are also given in Table 4.               

rating distribution across three distinct datasets, each collected from different real-world application areas (*Gulsoy, Yalcin & Bilge, 2023*).

This disproportionate focus on popular items creates a self-reinforcing cycle: the more a popular item is recommended and interacted with, the more likely it is to be recommended in the future. Consequently, less popular or new items, which might be highly relevant to certain user segments, remain obscured due to their lower initial interaction levels. This skewed distribution of user ratings leads to a narrow recommendation scope, heavily biased towards already popular items (*Abdollahpouri & Burke, 2019*). The challenge, therefore, lies in designing recommender systems that can recognize and correct for this imbalance, ensuring that the long tail of less popular items receives adequate attention (*Celma & Cano, 2008*). By addressing the core issue of imbalanced user rating distributions, recommender systems can move towards offering more diverse, inclusive, and personalized recommendations, breaking free from the constraints of popularity bias (*Abdollahpouri, Burke & Mobasher, 2019*, *2017*; *Borges & Stefanidis, 2021*).

The realm of popularity-debiasing methods in recommender systems can be broadly categorized into three distinct approaches: post-processing, in-processing, and pre-processing, each with its unique advantages and disadvantages (*Yalcin & Bilge, 2021*; *Boratto, Fenu & Marras, 2021*). Post-processing techniques are applied to the output of the recommendation algorithm. They typically involve re-ranking the recommended items to ensure a more balanced representation (*Abdollahpouri, Burke & Mobasher, 2019*;

*Gupta, Kaur & Jain, 2024*). However, this approach can sometimes be superficial, as it does not address the underlying bias in the model but merely adjusts its outputs. In-processing methods, on the other hand, integrate debiasing directly into the recommendation algorithm (*Kamishima et al., 2014*; *Hou, Pan & Liu, 2018*). This could involve modifying the algorithm to penalize the over-recommendation of popular items or explicitly boost diversity. While this approach can be more nuanced and directly tackle bias during the model training phase, it often requires significant alterations to existing algorithms and can be computationally intensive.

Pre-processing methods, which are central to the methodology proposed in this article, involve modifying the input data before it is fed into the recommendation algorithm (*Jannach et al., 2015*; *Chen et al., 2018*). The primary advantage of this approach is that it addresses the root cause of the bias—the skewed distribution of user interactions—by re-balancing or augmenting the data. This can lead to a more equitable representation of items, both popular and niche, within the system.

In this article, we introduce a novel pre-process popularity-debiasing method, namely the *EquiRate*, that focuses on strategically including synthetic ratings for less popular, or tail, items in the catalog. Our approach addresses two critical issues: firstly, determining which tail items to inject synthetic ratings into and calculating the optimal number of these ratings, and secondly, devising appropriate strategies for calculating the concrete value of these ratings in a way that does not disrupt the existing rating vectors of items.

To address the first issue, our method employs a data-driven strategy to assess the extent of imbalance in the user-item interaction data. Based on this assessment, we calculate an appropriate number of synthetic ratings needed to adequately represent the tail items, ensuring that their visibility in the recommender system is enhanced without overwhelming the dataset. For the second issue, the calculation of synthetic ratings is done with utmost care to maintain the authenticity and integrity of the original rating vectors. Our algorithm meticulously generates ratings that align with the underlying patterns and preferences reflected in the real user data. This ensures that the synthetic ratings blend seamlessly with the genuine ratings, thus preserving the natural dynamics of user-item interactions. By addressing these two issues, our proposed *EquiRate* method aims to strike a delicate balance between enhancing the representation of less popular items and maintaining the natural structure of the dataset. This approach not only mitigates popularity bias but also enriches the diversity and quality of recommendations, ultimately leading to a more balanced and user-centric recommender system.

The research questions (RQs) and main contributions of our article are presented below.

**RQ1:** *How can an innovative pre-processing debiasing method be designed to effectively balance training datasets by adjusting rating distributions among items, thereby mitigating the adverse effects of popularity bias in final recommendations?* We introduce a novel pre-processing technique, *i.e.*, the *EquiRate*, that adds synthetic

ratings to less popular (tail) items in the dataset. This method tackles the issue of popularity bias, enhancing the representation of these items for a balanced recommendation process.

**RQ2:** *How can a systematic approach be developed to determine the optimal allocation and quantity of synthetic ratings for tail items, ensuring the preservation of original rating vectors and natural user-item interaction patterns?* We present a systematic approach to determining the allocation of tail items to be injected into, and the optimal number and calculation of synthetic ratings. This ensures the integrity of the original rating vectors is maintained and the natural user-item interaction patterns are preserved.

**RQ3:** *How can rebalancing rating distributions between popular and less popular items improve diversity and fairness in recommendations, thereby enhancing user experience by increasing visibility for a broader range of items, including niche options?* By rebalancing the rating distribution between popular and less popular items, the proposed *EquiRate* method substantially improves the diversity and fairness of the recommendations. This enriches the user experience by increasing the visibility of a broader range of items, including niche but relevant options.

**RQ4:** *How can a holistic evaluation metric be developed to assess recommendation quality by harmonizing accuracy with beyond-accuracy dimensions, such as novelty, diversity, catalog coverage, and fairness?* We present the *FusionIndex*, an innovative metric for assessing the quality of recommendation lists that concurrently harmonizes accuracy with beyond-accuracy aspects, such as novelty, diversity, catalog coverage, and fairness, through a unique combination of multiple evaluation metrics.

The remaining sections of this study are organized as follows, respectively: (i) the section "Related Works" presents a detailed literature review regarding the existing popularity-debiasing strategies, (ii) the section "EquiRate: The Proposed Balanced Rating-Injection Strategy for the Popularity Bias Problem" introduces our proposed solution, the EquiRate, detailing its mechanism and effectiveness in mitigating popularity bias, (iii) the section "Experimental Studies" delves into the experimental setup, covering the datasets used, relevant parameters, and existing debiasing methods considered in the experiments and this section also elaborates on our evaluation metrics, focusing on the newly proposed the FusionIndex metric and presents the results of the performed experiments, comparing the performance of the EquiRate with other debiasing methods and highlighting its advantages, (iv) the section "Insights and Discussion" thoroughly discusses these findings and gives the most critical insights, and (v) the section "Conclusion and Future Work" concludes the article and suggests potential avenues for future research. Note that the source code accompanying this study is also made publicly available to enable the reproducibility of the experiment (https://github.com/SiriusFoundation/EquiRate, https://doi.org/10.5281/zenodo.12515959).

## RELATED WORKS

Recommender systems have become pervasive tools in individuals' everyday lives, offering personalized suggestions for commodities, amenities, and content. These systems heavily rely on data-driven algorithms to forecast user inclinations and generate recommendations. Nevertheless, there is a mounting apprehension regarding the susceptibility of these algorithms to popularity bias, whereby prevalent items are recommended more frequently compared to niche or less renowned alternatives (*Ahanger et al., 2022*). This bias has the potential to homogenize user preferences and restrict diversity in recommendations. In recent times, scholars have been concentrating their efforts on devising techniques to alleviate the popularity bias predicament in recommendation algorithms (*Chen et al., 2023*; *Yalcin, 2022*). This segment scrutinizes the extant literature pertaining to the issue of popularity bias and its corresponding remedies in recommender systems.

The preliminary investigations on this matter primarily examine the formulation and extent induced by recommendation algorithms in different fields, such as online education (*Boratto, Fenu & Marras, 2019*), movies (*Boratto, Fenu & Marras, 2021*; *Borges & Stefanidis, 2021*), books (*Naghiaei, Rahmani & Dehghan, 2022*), music (*Celma & Cano, 2008*; *Jannach, Kamehkhosh & Bonnin, 2016*; *Kowald, Schedl & Lex, 2020*), tourism (*Sánchez, 2019*), and social media (*Siino, La Cascia & Tinnirello, 2020*). One ground-breaking study comprehensively analyzes several prominent recommendation methods, particularly collaborative filtering algorithms, and reveals that their recommendations heavily favor a minuscule popular portion of the item spectrum (*Jannach et al., 2015*). This study also examines the impact of algorithmic design and parameterization on popularity bias and suggests hyperparameter tuning to enhance recommendation diversity. Another related work evaluates a set of representative algorithms against various biases associated with the popularity of course categories, catalog coverage, and course popularity in massive open online courses (*Boratto, Fenu & Marras, 2019*). It is concluded that the employed algorithms can differ significantly in the courses they recommend and may exhibit undesirable biases with corresponding educational implications. Moreover, another correlated investigation effectively evaluates the identical representative algorithms and illustrates the impact of the popularity bias issue on various entities, including individuals and item suppliers (*Abdollahpouri, 2020*).

Numerous recent investigations delve into the examination of the potential discrimination arising from the problem of popularity bias. In other words, these studies aim to explore how the bias perpetuated by recommendation algorithms towards popular items unjustly impacts various users with diverse characteristics or different stakeholders within the system, such as users or service providers (*Abdollahpouri, 2020*), or user classes based on age, race, or gender (*Lesota et al., 2021*). The initial study addressing the inequity of this matter observes that individuals who possess a primary interest in unpopular (*i.e.*, niche) items are more adversely affected by the popularity bias in movie recommendations (*Abdollahpouri et al., 2019*). This issue of unfairness among individuals has also been identified in the realms of books (*Naghiaei, Rahmani & Dehghan, 2022*) and

music recommendations (*Kowald, Schedl & Lex, 2020*). Moreover, this issue has been subjected to analysis with regards to five crucial attributes associated with users' rating behavior. The results indicate that individuals who are exacting, highly interactive, and difficult to predict face exceedingly unjust recommendations compared to others, despite their significant contributions to the system (*Yalcin & Bilge, 2022*). Furthermore, a recent study examines this issue of unfairness in relation to five fundamental personality traits of the big-five factor model. The study concludes that users who exhibit lower levels of extroversion or exhibit avoidance towards trying new experiences are subjected to more unjust referrals concerning item popularity (*Yalcin & Bilge, 2023*).

In the field of literature, numerous practical solutions have been devised to address the issue of bias propagation in recommender systems. These solutions are commonly classified into three categories: post-processing, in-processing, and pre-processing, based on their integration strategy during the recommendation generation phase.

Post-processing debiasing methods aim to re-order a ranked list generated by a conventional recommender or produce a new one by penalizing popular items while highlighting unpopular ones. These methods are widely used as they can be applied to the outputs of any recommender, making them more universally applicable than other methods. For instance, a study suggests the utilization of pre-defined user-specific weights to mitigate the issue of popularity bias (*Jannach et al., 2015*). These weights also enable the achievement of satisfactory accuracy performance. Another approach developed for this purpose involves calculating synthetic ranking scores by assigning weights to predictions inversely proportional to the popularity level of items (*Abdollahpouri, Burke & Mobasher, 2018*). Subsequently, items are arranged based on the ranking scores rather than the pure prediction values, resulting in the generation of final recommendation lists. In line with the strategy of penalizing popular items, we have previously proposed two enhanced re-ranking strategies, namely *Augmentative* and *Multiplicative*, to address the adverse effects of this issue in the domain of group recommendations (*Yalcin & Bilge, 2021*). Specifically, during the re-ranking process, the *Augmentative* approach places greater emphasis on ranking accuracy, while the *Multiplicative* approach prioritizes the inclusion of unpopular items in the recommendations. Lastly, a recent debiasing approach has been proposed to promote fairness in item exposure within recommendation systems by imposing constraints on the volume of stocks for items (*Dong, Xie & Li, 2021*). This approach restricts the maximum number of times an item can be recommended in proportion to its historical interaction frequency. Empirical validation has shown that this constraint outperforms conventional recommenders by effectively mitigating the Matthew Effect that influences item popularity. This heuristic approach relies on normalized scores and a minimum-cost maximum-flow strategy, utilizing both the existing recommendation length of users and our item stock volumes. As a result, both the problem of user-item matching and the issue of popularity bias in the final recommendations are effectively addressed.

Additionally, an influential research study presents a highly efficient long-tail promotional technique inspired by the *xQuad* algorithm in order to enhance the quality of diversity in recommendations (*Abdollahpouri, Burke & Mobasher, 2019*). In another

recent approach to mitigate popularity bias, discrepancy minimization aims to amplify aggregate diversity in recommendations through a minimum-cost network flow strategy, thereby exploring recommendation sub-graphs that optimize diversity (*Antikacioglu & Ravi, 2017*). Conversely, an alternative technique for debiasing popularity, known as FA*IR, concentrates on achieving a balance between popular and unpopular items by establishing queues for these two categories and subsequently merging them (*Zehlike et al., 2017*). Lastly, to the best of our knowledge, in the existing literature, Calibrated Popularity stands as the sole method that takes into account users' original preferences towards item popularity during debiasing (*Abdollahpouri et al., 2021*). This method initially determines users' genuine interests across three categories of items, namely head, middle, and tail, which are constructed based on the number of received ratings. Subsequently, this information is integrated into the Jensen-Shannon divergence when generating new recommendation lists based on rankings. However, the primary limitation of this approach lies in the fact that it considers all ratings in users' profiles without considering their values when determining users' interests in popularity. It should be noted that users have the capability of providing numerous highly negative votes for popular items, and assigning a rating to an item by a user does not necessarily imply their enjoyment of said item.

In-processing popularity-debiasing methods, on the other hand, have the objective of modifying the internal mechanism of a recommendation algorithm in order to counteract the bias it may have towards item popularity. These methods are commonly known as non-generalizable solutions, as they only provide assistance to specific recommenders and cannot be applied universally. One instance of such methods involves evaluating the likelihood of individuals disliking certain items and utilizing this information to penalize popular items during the estimation of recommendations (*Kamishima et al., 2014*). Furthermore, a recent study introduces an optimized variation of the well-known RankALS recommender that aims to achieve recommendation lists with improved intra-list diversity and ranking accuracy (*Abdollahpouri, Burke & Mobasher, 2017*). In addition, another related work presents a recommendation framework that initially estimates the common neighbors of two items based on their initial popularity and subsequently removes the most popular ones to attain a more balanced co-neighbor similarity index (*Hou, Pan & Liu, 2018*). Moreover, another approach focuses on debiasing popularity by minimizing the correlation between the relevance of user-item pairs and item popularity, resulting in a more equitable treatment of items in the long tail (*Boratto, Fenu & Marras, 2021*). The final method in this category is founded on variational autoencoders, which penalizes the scores assigned to items based on their historical popularity to mitigate bias and enhance diversity in the recommended results (*Borges & Stefanidis, 2021*). Finally, a novel paradigm named Popularity-bias Deconfounding and Adjusting (PDA) is proposed in *Zhang et al. (2021)*, aiming to remove confounding popularity bias in model training and adjust recommendation scores with desired popularity bias during inference.

The prevalence of bias in recommendations is often attributed to the unequal distribution of ratings provided by users. Therefore, pre-processing methods aim to mitigate this imbalance by modifying the data on which the recommender systems are

trained. For instance, one pre-processing approach involves categorizing items in the catalog as either "head" or "tail" based on their received ratings. Predictions are then generated using all ratings for head items, while ratings from the corresponding class are used for tail items (*Park & Tuzhilin, 2008*). This ensures that the predictions for tail items are not influenced by the dominance of popular items. Another approach in this category involves constructing user-item tuples, where popular items are excluded. The recommendation algorithms are subsequently trained on these tuples instead of the original user-item rating matrix (*Jannach et al., 2015*). Lastly, another approach involves utilizing a probability distribution function that considers the popularity level of items. This allows for a higher likelihood of tail items being included in the ranking-based recommendations (*Chen et al., 2018*).

*Zhou et al. (2023)* introduce an efficient method that dynamically down-weights graph neighbours according to item-level popularity, thereby attenuating bias within the embedding propagation itself. Complementing this, *Lopes et al. (2024)* propose a post-processing framework that enforces minimum exposure guarantees: each item is assigned a lower-bound on display frequency, and a constrained optimisation routine re-ranks the baseline list so that these guarantees are satisfied while relevance loss is minimised. At the re-ranking layer, *Naghiaei, Rahmani & Deldjoo (2022)* present CPFair, which jointly personalises fairness for both consumers and producers by balancing individual user utility with long-tail exposure. Collectively, these studies underscore the community's shift toward joint accuracy–fairness objectives and sit orthogonally to our pre-processing paradigm: *EquiRate* can be coupled with such in-or post-processing schemes to furnish a multi-stage defence against popularity bias throughout the recommendation pipeline.

A recent study addresses the problems of unfairness and popularity bias in recommendation systems by considering users' interactions with popular items. The proposed algorithm reduces popularity bias and unfairness while slightly increasing recommendation accuracy by eliminating these so-called "unreliable interactions" (*Ma & Dong, 2021*). The model proposed in another related work aims to reduce recommendation bias and ensure recommendation utility in recommendation systems by addressing how users' ratings are influenced by herd mentality (*Su, Li & Zhu, 2023*). Also, a recent work presents an algorithm to reduce the popularity bias in recommender systems (*Gangwar & Jain, 2021*). This bias tends to over-recommend popular items while ignoring non-popular ones. The proposed algorithm adjusts the weights of non-popular items, improving their representation. Extensive testing demonstrates that this method effectively reduces popularity bias while maintaining recommendation accuracy. Finally, the recently introduced *TASTE* model uses text representations to match items and users, enhancing recommendation accuracy (*Liu et al., 2023*). The *TASTE* effectively reduces popularity bias and addresses the cold start problem by leveraging full-text modeling and pre-trained language models. This method results in more relevant and appropriate recommendations.

As can be inferred from the aforementioned literature review, numerous efforts have been made to address the concern of popularity bias. For instance, post-processing techniques, which intervene after the recommender system has made its predictions, can

lead to decreased accuracy and increased system complexity. Moreover, these methods can sometimes be superficial, as they do not address the underlying bias in the model but merely adjust its outputs. In contrast, in-processing methods, which are integrated during the model's training phase, can increase the complexity and training duration of the model, and may alter the data structure. However, pre-processing approaches can avoid these issues by rectifying biases in the dataset before the training process of the recommender system begins. These methods work on the data prior to the training phase, thereby reducing bias without increasing model complexity and maintaining the system's accuracy. Hence, pre-processing methods can offer a more effective and efficient solution to address popularity bias in recommendation systems compared to post-processing and in-processing methods.

Furthermore, existing pre-processing solutions for the popularity bias problem are often evaluated based on a limited set of criteria, typically focusing on accuracy or the level of popularity bias. However, there are numerous aspects beyond accuracy that influence the quality of recommendations, including *novelty*, diversity, *entropy*, and *long-tail coverage*, all of which can be impacted by popularity bias. Consequently, more robust approaches are needed to simultaneously address these criteria. Our proposed method, the *EquiRate*, meets this need by not only balancing the rating distribution across items through the injection of ratings that align with patterns and preferences in real user data, but also by generating recommendation lists that meet various criteria, such as accuracy and other beyond-accuracy aspects, concurrently.

## EQUIRATE: THE PROPOSED BALANCED RATING INJECTION STRATEGY FOR THE POPULARITY BIAS PROBLEM

Popularity bias in recommender systems is a pervasive issue that stems from inherent imbalances in data distribution. Typically, a small fraction of items, often the most popular or 'head' items, receive a disproportionately large amount of user interactions and ratings. This leaves a long tail of less popular, or 'tail', items that are seldom recommended. Such an imbalanced distribution leads to a feedback loop, where popular items gain more visibility and become even more dominant, while tail items remain obscured. This not only limits the diversity of recommendations provided to users but also hinders the discovery of new or niche content, ultimately affecting the user experience and the system's effectiveness (*Boratto, Fenu & Marras, 2021*; *Gulsoy, Yalcin & Bilge, 2023*).

The root cause of this bias lies in the way recommender systems are traditionally designed. Most algorithms are geared towards optimizing accuracy based on existing user-item interactions, inadvertently favoring items with higher numbers of interactions. This leads to a situation where the rich get richer—popular items get more recommendations and hence more ratings, further reinforcing their popularity. This skewed emphasis on popular items can be detrimental, as it narrows the breadth of content surfaced to users, often overlooking potentially relevant but less popular items.

To tackle this challenge, our proposed method, the *EquiRate*, introduces a novel approach to recalibrate the recommendation landscape. The *EquiRate* is specifically designed to mitigate the effects of popularity bias by injecting synthetic ratings into tail items. This method aims to balance the scales, giving these less popular items a fair chance to be recommended and noticed by users.

The *EquiRate* operates on the principle of identifying and enhancing the visibility of tail items in a dataset. While some clustering-based methods exist for classifying items into head and tail categories (*Park & Tuzhilin, 2008*), we adopt the Pareto principle (*Sanders, 1987*), as it is more widely recognized and frequently used in the literature on popularity bias (*Abdollahpouri, Burke & Mobasher, 2019*; *Yalcin & Bilge, 2023*; *Gulsoy, Yalcin & Bilge, 2023*). The method then involves the strategic addition of synthetic ratings to these tail items, calculated through a series of carefully devised algorithms and formulas. These synthetic ratings are not arbitrary but are thoughtfully computed to reflect realistic user preferences, thereby maintaining the authenticity of the recommender system. By injecting synthetic ratings in a controlled and measured way, the *EquiRate* disrupts the feedback loop that perpetuates the dominance of popular items. This intervention allows for a more diverse and representative item selection in recommendation lists. As a result, users can get a chance to interact with a broader range of products, services, or content, which they might not have encountered otherwise due to the overshadowing presence of popular items.

In summary, the *EquiRate* presents a strategic solution to address the skewness in the data distribution that underpins popularity bias in recommender systems. It does this by enhancing the visibility and likelihood of recommendation for less popular items, thereby creating a more balanced, diverse, and representative recommendation ecosystem. This approach not only enriches the user experience by broadening the range of recommendations but also ensures a more equitable platform for all items, regardless of their initial popularity. The ultimate goal of the *EquiRate* is to foster a recommendation environment where items are judged and recommended based on their relevance and quality, rather than solely on their existing popularity metrics.

More concretely, the first step in the *EquiRate* involves classifying items into different popularity classes using the Pareto principle. Let $I$ denote the set of all items, and let $R_i$ be the number of ratings received by item $i$. The items are then sorted in descending order based on $R_i$. The first $M$ items receiving 20% of all ratings in the system are categorized as 'popular' or head items, denoted by $H$, while the remaining ones received the remaining 80% of the ratings are classified as tail items, denoted by $T$. The *EquiRate* employs three distinct strategies, **Synthetic Rating Injection (SRI)** strategies, for calculating the number of synthetic ratings to inject into each tail item. These strategies are explained in detail in the following:

***Overall Popularity Adjustment (OPA):*** this strategy first calculates the popularity score ($Pop_i$) for each item $i$ based on the number of ratings it has received and the average popularity score ($Pop_{avg}$) for all items. Then, the *OPA* strategy determines the number of

synthetic ratings to allocate to each tail item ($n_i$), as in Eq. (1), where $\alpha$ is a scaling factor in the range [0, 1], adjusting the total number of ratings to be injected.

$$n_i = \alpha \times (Pop_{avg} - Pop_i). \tag{1}$$

***Head Item-Focused Adjustment (HIFA):*** similar to the *OPA*, this strategy first computes $Pop_i$ for each item *i*, but differing from the *OPA*, the *HIFA* strategy calculates the average popularity score ($Pop_{head\_avg}$) for only head items. Then, it determines the number of synthetic ratings to allocate to each tail item ($n_i$), as in Eq. (2), where $\alpha$ is again the scaling factor in the range [0, 1].

$$n_i = \alpha \times (Pop_{head\_avg} - Pop_i). \tag{2}$$

***Threshold-based Popularity Adjustment (TPA):*** similar to other strategies, it first calculates $Pop_i$ for each item *i* in the catalog; however, the *TPA* strategy sets a popularity threshold $\theta$ based on the lowest popularity among head items, and then it determines the number of synthetic ratings to allocate to each tail item ($n_i$), as in Eq. (3), where $\alpha$ is again the scaling factor in the range [0, 1].

$$n_i = \alpha \times (\theta - Pop_i). \tag{3}$$

After determining the quantity of fake ratings to be injected, the next crucial step is to decide how to assign rating values to the empty cells within the profiles of tail items. To address this, we employ various ***Synthetic Rating Generation (SRG)*** strategies outlined below. These strategies ensure that the synthetic ratings are introduced in a manner that maintains the integrity of item profiles while reducing popularity bias. To complete the *EquiRate* pipeline, once the number of synthetic ratings to be injected into each tail item is determined *via* a selected *SRI* strategy, and their values are computed using an *SRG* strategy, the next critical component is identifying which users should be assigned these synthetic interactions. In our framework, user selection is conducted through random sampling from the pool of users who have not previously interacted with the given tail item. This design choice serves two primary purposes: (*i*) it preserves the statistical independence of synthetic ratings by avoiding systemic user-item correlations, and (*ii*) it enables broad coverage across the user base, which improves exposure of tail items in the overall matrix without biasing toward specific users. While more complex user-targeting schemes could be employed (*e.g.*, based on user similarity or predictive matching), our experiments showed that random user selection offers sufficient fairness and diversity while maintaining simplicity and computational efficiency. This mechanism ensures that each synthetic rating maintains the intended effect of breaking popularity feedback loops without introducing new bias into the dataset.

***Generalized Rating Value (GRV):*** in this strategy, we calculate the values of fake ratings based on general statistics derived from both item and user profiles. These statistics may include the average rating of all items, the average rating provided by the respective user, or the average rating received by the item itself.

***Predictive Rating Value (PRV)***: here, we opt for a more personalized approach by generating synthetic ratings for tail items using recommendation algorithms, such as collaborative filtering ones. These algorithms are initially trained on the original user-item matrix and are then applied to create synthetic ratings that are specific to each user and item.

***Randomized Value Generation (RVG)***: it is inspired by privacy-preserving collaborative filtering techniques (*Polat & Du, 2005*; *Bilge & Polat, 2013*), which aim to safeguard user privacy in recommender systems. In doing so, we adopt randomized perturbation (*Gulsoy, Yalcin & Bilge, 2023*), one of the most utilized approaches to perturb each genuine rating randomly. Accordingly, randomly generated rating vectors will be added to the empty cells of the actual rating vectors of tail items. Similar to the original randomized perturbation approach, these vectors will be generated using either a uniform or normal distribution. In the uniform distribution, random number values are generated within the range of $[-\sqrt{3}\sigma, \sqrt{3}\sigma]$, while in the normal distribution, random number values follow the $N(\mu, \sigma^2)$ distribution. Notably, the average of the random number values generated in both distributions is zero.

In summary, the procedures of the *EquiRate* popularity-debiasing method are outlined in Algorithm 1. This method can be implemented in nine distinct variants by choosing one of three different approaches from both the *SRI* and *SRG* categories. In the subsequent experiments of our study, we aim to identify which combination most effectively achieves popularity-debiasing. Once the optimal *EquiRate* variant is determined, it can be used to preprocess a dataset for any recommendation algorithm, such as those based on collaborative filtering. This preprocessing results in a balanced rating dataset (denoted as $R'$). Utilizing this balanced dataset, rather than the original, for recommendation algorithms, is likely to improve the representation of less popular (*i.e.*, tail) items in the final recommendations. Consequently, this enhances beyond-accuracy qualities such as diversity, novelty, and fairness in the item recommendations. We also present an overview of the general structure of the proposed *EquiRate* popularity-debiasing method in Fig. 2.

To reassure practitioners that *EquiRate* can be integrated into existing pipelines with negligible overhead, we analyse its worst-case time and space requirements. Let $|T|$ be the number of tail items, $k_i$ the number of synthetic ratings injected into item $i$, and $|U|$ the total number of users. The SRI step costs only $O(|T|)$ to compute $\{k_i\}$, because it involves simple arithmetic over pre-computed popularity counts. Selecting user–item pairs to receive injections is $O(\sum_{i \in T} k_i)$, as each pair is sampled exactly once. Rating-value generation is likewise lightweight: GRV requires an $O(1)$ table look-up per pair; PRV adds a single forward pass of a pre-trained recommender, leaving overall complexity unchanged relative to baseline training; RVG merely draws one normal (or uniform) random variate per pair, also $O(1)$. Consequently, the entire pre-processing pipeline is *linear* in the number of injected ratings and does not alter the asymptotic cost of subsequent model training. Memory overhead follows the same bound, $O(\sum_{i \in T} k_i)$, because synthetic entries are stored in the same sparse format as genuine ratings.

---

**Algorithm 1** *EquiRate*: balanced rating injection strategy.

**Input:** User-Item Rating Matrix $R$,
SRI Strategy $\mathscr{S}_{\text{SRI}}$ (OPA, HIFA, or TPA),
SRG Strategy $\mathscr{S}_{\text{SRG}}$ (GRV, PRV, or RVG),
Scaling factor $\alpha$
**Output:** Augmented Matrix $R'$
**Step 1: Popularity Classification using Pareto principle**
Compute rating counts $R_i$ for all items $i \in I$
Sort items in descending order of $R_i$
Determine head items $H$ covering top 20% of all ratings
Define tail item set $T = I \setminus H$
**Step 2: Compute Synthetic Rating Counts**
**foreach** *tail item* $i \in T$ **do**
    Compute popularity score $Pop_i$ based on $R_i$
    **if** $\mathscr{S}_{\text{SRI}}$ *is OPA* **then**
        Compute $Pop_{avg}$ for all items
        $n_i \leftarrow \alpha \times (Pop_{avg} - Pop_i)$          ▷ *using Eq. (1)*
    **else if** $\mathscr{S}_{\text{SRI}}$ *is HIFA* **then**
        Compute $Pop_{head\_avg}$ for items in $H$
        $n_i \leftarrow \alpha \times (Pop_{head\_avg} - Pop_i)$      ▷ *using Eq. (2)*
    **else if** $\mathscr{S}_{\text{SRI}}$ *is TPA* **then**
        Determine threshold $\theta = \min_{j \in H}(Pop_j)$
        $n_i \leftarrow \alpha \times (\theta - Pop_i)$            ▷ *using Eq. (3)*
**Step 3: Inject Synthetic Ratings *via* Random User Assignment**
**foreach** *tail item* $i \in T$ **do**
    Randomly sample $n_i$ users from $U$ who have not rated item $i$
    **foreach** user $u$ in sampled users **do**
        **if** $\mathscr{S}_{\text{SRG}}$ *is GRV* **then**
            Compute $r_{ui}$ using global/user/item mean
        **else if** $\mathscr{S}_{\text{SRG}}$ *is PRV* **then**
            Predict $r_{ui}$ using any trained CF recommendation model
        **else if** $\mathscr{S}_{\text{SRG}}$ *is RVG* **then**
            Generate $r_{ui}$ *via* random noise (uniform or normal)
        Inject synthetic rating $r_{ui}$ into $R'$
**return** $R'$

---

## A toy example illustrating all *EquiRate* variants

To clearly demonstrate the inner workings of the proposed *EquiRate* framework, this subsection presents a toy example that operationalizes all six possible variants resulting from the combination of three *Synthetic Rating Injection (SRI)* strategies and three *Synthetic Rating Generation (SRG)* strategies. This example simulates a simplified

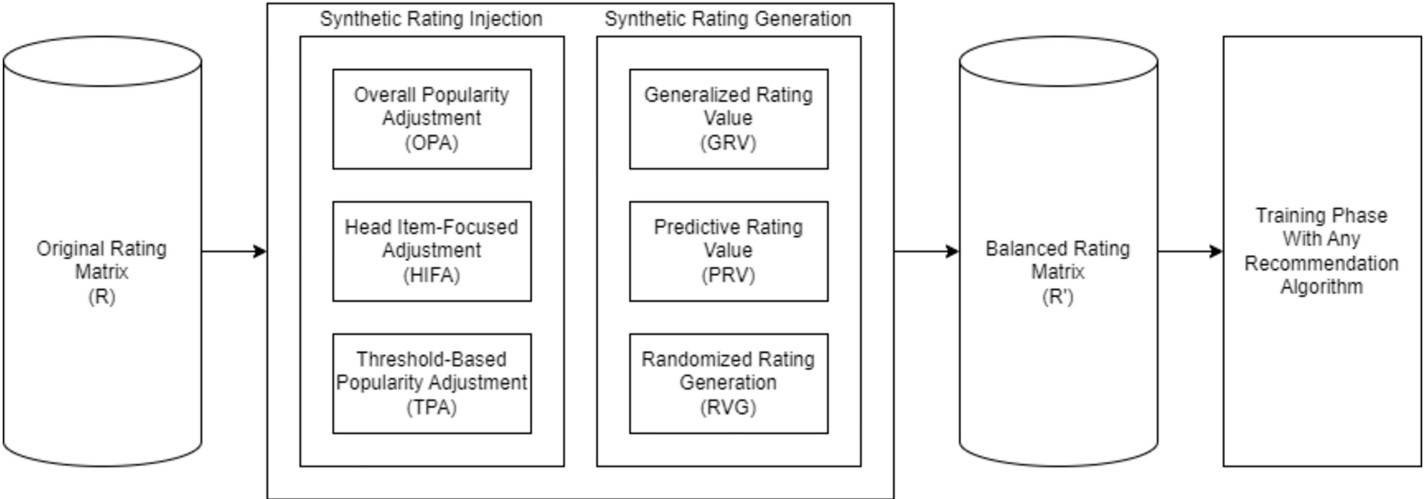

**Figure 2** General structure of the proposed *EquiRate* approach.               

**Table 1  Initial user-item rating matrix.**

| User | A | B | C | D | E |
|------|---|---|---|---|---|
| $U_1$ | 4 | 5 | – | – | – |
| $U_2$ | 5 | – | 3 | – | 2 |
| $U_3$ | 4 | 1 | 5 | 1 | – |
| $U_4$ | 3 | – | 4 | 5 | – |
| $U_5$ | 2 | – | – | 3 | – |
| $U_6$ | 5 | – | – | – | – |
| $U_7$ | 4 | – | – | – | – |

user-item rating matrix and walks through the entire pipeline of the *EquiRate* algorithm, from identifying tail items to injecting synthetic ratings.

**Step 1: Initial Matrix and Popularity Classification:** Table 1 shows a simplified user-item rating matrix with seven users ($U_1$ to $U_7$) and five items ($A$ to $E$). Each rating is on a scale from 1 to 5. The matrix is sparse and reflects the typical long-tail distribution, where a small number of items receive a large portion of ratings. Specifically, item $A$ receives seven ratings, $C$ and $D$ receive three each, $B$ receives two, and $E$ receives only one.

Using the Pareto principle, we identify head and tail items. In this case, the total number of ratings is 16. The first 20% of ratings (*i.e.*, 3.2 ratings) fall entirely on item $A$, which is therefore classified as the sole head item. The remaining items $B$, $C$, $D$, and $E$ are categorized as tail items.

**Step 2: Calculating Number of Synthetic Ratings (SRI):** we now apply each of the three SRI strategies to determine the number of synthetic ratings to inject into each tail item. Assume a scaling factor of $\alpha = 1$ for simplicity.

- **Overall Popularity Adjustment (OPA):** the average popularity across all items is $16/5 = 3.2$. For each tail item $i$, the number of synthetic ratings $n_i = 3.2 - Pop_i$ where $Pop_i$ is the number ratings item $i$ has received, as formulated in Eq. (1). Accordingly, the number of synthetic ratings to be injected for $B$ is $3.2 - 2 = 1.2$, for $C$ is $3.2 - 3 = 0.2$, for $D$ is $3.2 - 3 = 0.2$, and for $E$ is $3.2 - 1 = 2.2$.

- **Head Item-Focused Adjustment (HIFA):** the average popularity of head items is 7 (only item $A$). Then, for each item in the tail set, the number of synthetic ratings $n_i$ can be computed as $7\text{-}Pop_i$ where $Pop_i$ is the number of ratings item $i$ has received (see Eq. (2)). Accordingly, the number of ratings to be included with HIFA for $B$ is $7 - 2 = 5$, for $C$ is $7 - 3 = 4$, for $D$ is $7 - 3 = 4$, and for $E$ is $7 - 1 = 6$.

- **Threshold-based Popularity Adjustment (TPA):** in the given toy example, the head item set contains only one item ($A$), which received seven ratings. Since the popularity threshold ($\theta$) in the TPA strategy is defined as the lowest popularity score among head items, and there is only a single head item in this case, the threshold becomes equal to the popularity of item $A$. As a result, the TPA strategy essentially mirrors the HIFA strategy in this specific setting, as both rely on the same reference value to compute synthetic ratings for tail items.

In summary, the number of synthetic ratings to be injected into each tail item using the three SRI strategies is presented in Table 2. It is noteworthy that in this toy example, both TPA and HIFA produce identical results because the head item set consists of only a single item. However, in real-world datasets with multiple head items, the threshold used in TPA —defined as the minimum popularity among head items, which is typically lower than the HIFA average. Consequently, TPA usually results in fewer injected ratings than HIFA, rendering it a more conservative adjustment strategy. Furthermore, fractional values produced during the computation (*e.g.*, 1.2 or 0.2) are rounded to the nearest integer to facilitate practical implementation. The values reported in Table 2 reflect this rounding.

**Step 3: Generating Synthetic Values (SRG):** once the number of synthetic ratings is determined, the next step is to assign rating values to user-item pairs. Three SRG strategies are employed for this:

- **Generalized Rating Value (GRV):** in the GRV strategy, synthetic rating values are derived from aggregate statistics, such as the global average rating, the average rating provided by a user, or the average rating received by an item. The user-item pairs selected for injection are randomly chosen from the set of unrated tail items. For instance, based on Table 2, if the *HIFA* strategy is selected for the SRI component, item $E$ would receive six synthetic ratings. Suppose user $U_4$ is randomly assigned to one of these six injections and their average rating is 4.0. Then, the injected rating value would also be 4.0.

- **Predictive Rating Value (PRV):** in the PRV strategy, synthetic rating values are predicted using a trained recommender algorithm (*e.g.*, matrix factorization or user-based collaborative filtering). After determining the number of synthetic ratings to be added to each tail item—*e.g.*, six for item $E$ under *HIFA*—the system randomly

**Table 2 Number of synthetic ratings to be injected per tail item (SRI Strategies).**

| Item | OPA | HIFA | TPA | # Of actual ratings |
|------|-----|------|-----|---------------------|
| B | 1 | 5 | 5 | 2 |
| C | 0 | 4 | 4 | 3 |
| D | 0 | 4 | 4 | 3 |
| E | 2 | 6 | 6 | 1 |

**Table 3 An example of final user–item matrix after applying the *OPA* (SRI) and *GRV–Users' Mean* (SRG) variant.**

| User | A | B | C | D | E |
|------|---|---|---|---|---|
| $U_1$ | 4 | 5 | – | – | $5^S$ |
| $U_2$ | 5 | – | 3 | – | 2 |
| $U_3$ | 4 | 1 | 5 | 1 | $3^S$ |
| $U_4$ | 3 | $4^S$ | 4 | 5 | – |
| $U_5$ | 2 | – | – | 3 | – |
| $U_6$ | 5 | – | – | – | – |
| $U_7$ | 4 | – | – | – | – |

**Note:**
Italic entries with superscript *S* denote injected synthetic ratings.

selects users who have not rated that item. For each selected user-item pair, the recommender model is used to predict the rating, which is then injected into the user-item matrix.

○ **Randomized Value Generation (RVG):** this strategy generates synthetic ratings by adding noise drawn from a normal distribution, following the number of injections computed by the chosen SRI strategy (*e.g.*, Six ratings for item *E* under HIFA, see Table 2). For each selected user-item pair (chosen randomly from unrated entries), a noise value ε is derived from $N(0, \sigma^2_{\max})$ and added to the user's average rating. Here, we adopt three noise levels: *Low* ($\sigma_{\max} = 2$), *Mid* ($\sigma_{\max} = 3$), and *High* ($\sigma_{\max} = 4$), following *Gulsoy, Yalcin & Bilge (2023)*. The computed values are clipped to fit the rating scale if necessary.

Table 3 illustrates an example of matrix densification achieved by the *EquiRate* pipeline when the OPA rule is coupled with GRV based on users' mean ratings. Consistent with Table 2, exactly one synthetic rating was injected into item *B* ($U_4 = 4$) and two into item *E* ($U_1 = 5$, $U_3 = 3$); no other cells were altered. These values equal the rounded averages of the respective users' existing ratings, thereby preserving user-level preference profiles while selectively boosting feedback for under-represented items. The result is a modest yet targeted reduction of popularity imbalance that subsequent recommendation models can exploit without distorting the overall rating distribution.

This example illustrates the complete execution of the *EquiRate* framework: tail items are first identified, synthetic-rating quantities are computed *via* the three SRI rules, and the

resulting blanks are then filled with values generated by the SRG strategies. Because each SRI rule can be paired with each SRG rule, nine distinct variants arise, offering practitioners ample flexibility to match the variant to a specific dataset and fairness objective. In the empirical study that follows, all nine combinations are exhaustively evaluated on three benchmark datasets; the analysis then focuses on the variants that deliver the best trade-off between accuracy and beyond-accuracy metrics. This concrete, numerical walk-through therefore not only clarifies the mechanics of *EquiRate* but also underpins the transparency and replicability of the subsequent experimental findings.

## EXPERIMENTAL STUDIES

This section provides detailed information about the datasets used, the parameter setup for the proposed *EquiRate* method, and prominent popularity-debiasing methods for comparison. Additionally, it introduces an advanced evaluation metric that considers both accuracy and beyond-accuracy aspects of the recommendations concurrently. The section also presents the results obtained from the experiments conducted.

### Datasets

Our study incorporates three distinct public benchmark datasets from various application realms, as employed by highly related recent studies (*Yalcin & Bilge, 2022*; *Gulsoy, Yalcin & Bilge, 2023*). These include the MovieLens-1M (https://grouplens.org/datasets/movielens/1m/) (ML) dataset for movies (*Harper & Konstan, 2015*), the Douban Book (https://www.douban.com/) (DB) dataset for books (*Shi et al., 2018*), and the Yelp (https://www.yelp.com/) dataset for local business reviews (*Shi et al., 2018*). In each dataset, user preferences are represented as discrete values on a five-star scale. Table 4 offers an in-depth view of the ML, DB, and Yelp datasets. Their varying dimensions and sparsity levels, as shown in Table 4.

### Parameter setup

As discussed previously, two critical points related to the experiments are determining how many synthetic ratings to add to which items using any SRI method and then calculating the value of ratings to be added *via* any SRG method.

However, the number of scenarios in the experiments will be more efficient in exploring the parameter options where the proposed method can perform best. Therefore, nine main scenarios to be run with a combination of three SRI methods (*i.e.*, OPA, HIFA, and TPA) and three SRG methods (*i.e.*, GRV, PRV, and RVG) are diversified to 156 different scenarios with different parameter values. Accordingly, the $\alpha$ parameter controlling the rating density to be injected with the SRI methods is planned with eight values varying from 0 to 1 (*i.e.*, 0.1, 0.25, 0.33, 0.50, 0.66, 0.75, 0.90, and 1.00). Also, synthetic rating calculation options with SRG methods are planned using three different methods. However, GRV methods are tested with three sub-approaches (the average ratings of a corresponding user (*i.e.*, $u_{avg}$), the average rating of a corresponding item (*i.e.*, $i_{avg}$), and the

**Table 4 Detailed overview of the MovieLens-1M (ML), Douban book (DB), and yelp datasets.**

| Dataset | Ratings | Sparsity (%) | Average rating value | Users | Items | Ratings per user | Ratings per item |
|---|---|---|---|---|---|---|---|
| ML | 1,000,209 | 95.8 | 3.58 | 6,040 | 3,952 | 165.6 | 253.1 |
| DB | 792,062 | 99.7 | 4.05 | 13,024 | 22,347 | 60.8 | 35.4 |
| Yelp | 198,397 | 99.9 | 3.77 | 16,239 | 14,284 | 12.2 | 13.9 |

average of all ratings in the dataset (*i.e.*, $all_{avg}$)). On the other hand, when applying the PRV method, the VAECF algorithm is used as it is one of the best-performing and up-to-date collaborative filtering approaches in terms of accuracy in the literature (*Liang et al., 2018*). Finally, the RVG method is considered with three scenarios where the number of ratings injected varies by adjusting privacy parameters: *Low*, *Middle (Mid)*, and *High*. In doing so, when applying the RPT method, $\sigma$ values are uniformly randomly selected from $(0, \sigma_{max}]$ interval where $\sigma_{max}$ is set 2, 3, and 4 for *Low*, *Mid*, and *High* scenarios, respectively. Note that these parameters are selected as in *Gulsoy, Yalcin & Bilge (2023)* and we do not observe significant differences between uniform and normal distribution in the experiments performed for the RVG method, therefore, we only present the outcomes that are obtained *via* normal distribution.

In summary, combining three SRI approaches, eight different $\alpha$ parameters, and seven SRG approaches resulted in 156 experimental scenarios. However, even if all scenarios are tested in the experiments, for clarity, we only present outcomes of the best-performing variants of the *EquiRate* in the following section. Here, we present the results where $\alpha$ is selected as 0.1, 0.5, and 0.9 to analyze better its effect on the quality of final recommendations. These variants are repeated five times to obtain reliable outcomes. Then, we take the average of these repetitions to make the randomness more reliable and the values more stable. Figure 3 also sketches all tested parameter-tuning for the proposed *EquiRate* method. In the final stage of our study, we utilize the VAECF algorithm on the dataset, which has been balanced by applying our *EquiRate* variants. This process aims to generate top-*N* recommendation lists for each individual, where we have chosen *N* to be 10. It should also be noted that the implementations of the VAECF algorithm are carried out using the recently developed Python-based framework known as Cornac (https://cornac.preferred.ai/) (*Salah, Truong & Lauw, 2020*).

During the experiments, we adopt a user-based leave-one-out cross-validation scheme, a common protocol for top-*N* recommendation studies. For each fold, one user *u* is held out as the test (*i.e.*, active) user while all remaining users constitute the training set. The recommendation model (VAECF in all experiments) is trained on this set and then used to predict scores for every item in the catalogue for *u*. Then, the top-10 items with the highest predicted scores form the recommendation list. We repeat this procedure for every user, so that each user acts exactly once as a test, and $|U|$ folds are produced. Except for *Entropy* and *LTC*, all evaluation metrics are computed per user and averaged. Since these two metrics are system-level, we first merged the top-10 recommended items to obtain a list that includes all recommended items in the system to compute *Entropy* and *LTC*.

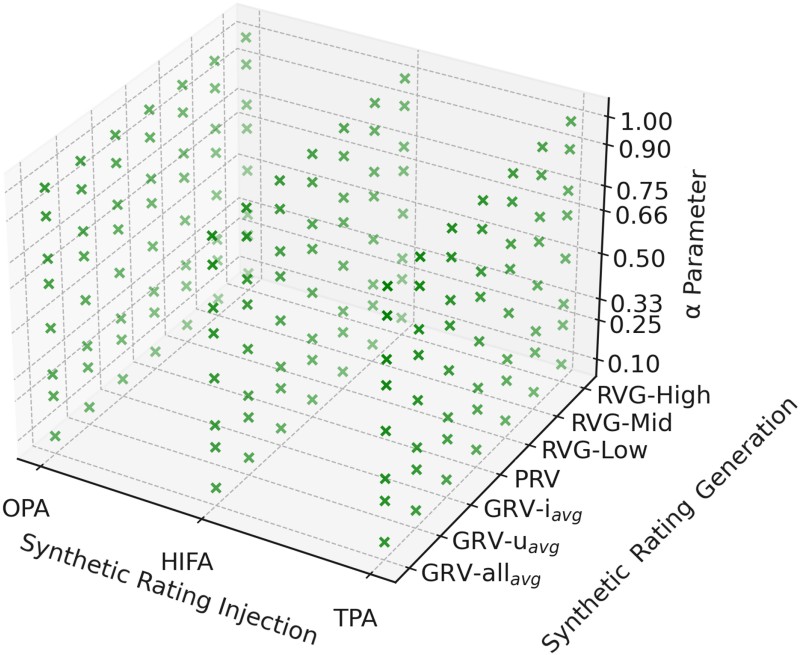

**Figure 3** **All tested parameter-tuning for the *EquiRate*.** Green dots representatively show each variant of the *EquiRate*.

## Benchmark popularity-debiasing methods

In our study, we compare our method with four prominent strategies to mitigate popularity bias: *eXplicit Query Aspect Diversification (xQuad)* (*Abdollahpouri, Burke & Mobasher, 2019*), *Augmentative (AUG)* (*Yalcin & Bilge, 2021*), *Popularity-aware Weighting (PAW)* (*Abdollahpouri, Burke & Mobasher, 2018*), *Multiplicative (MUL)* (*Yalcin & Bilge, 2021*), *Largest-Normalized-Score-First (LNSF)* (*Dong, Xie & Li, 2021*), *CP* (*Abdollahpouri et al., 2021*), *DM* (*Antikacioglu & Ravi, 2017*), *FA\*IR* (*Zehlike et al., 2017*), and *EqBal-RS* (*Gupta, Kaur & Jain, 2024*). The following explains these methods in detail:

1. *EXplicit Query Aspect Diversification (xQuad) (Abdollahpouri, Burke & Mobasher, 2019)*: this approach, a well-known method for reducing popularity bias, re-ranks items into two categories: popular (*i.e.*, head) and less popular (*i.e.*, tail). It aims to diversify final recommendations by balancing these categories, based on user preferences. A controlling parameter, $\lambda$, set at 0.5, helps balance accuracy and diversity.

2. *Augmentative (AUG) (Yalcin & Bilge, 2021)*: the *AUG* combines pure prediction scores with inverse weights of item popularity to create synthetic ranking scores. Prediction scores are the primary factor, while item weights serve as an additive element, emphasizing ranking accuracy over popularity bias mitigation.

3. *Popularity-aware Weighting (PAW) (Abdollahpouri, Burke & Mobasher, 2018)*: this method calculates logarithmic inverse weights for items based on popularity, giving more weight to less popular items and less to more popular ones. It then integrates these

weights with prediction scores to derive ranking scores for final recommendations. A parameter, $\lambda$, set at 0.5, balances these two recommendation aspects.

4. *Multiplicative (MUL) (Yalcin & Bilge, 2021):* similar to the *AUG* in using a re-ranking strategy that penalizes popular items, the *MUL*'s approach is distinct. It applies item weights as a multiplicative factor in calculating final ranking scores, aiming to favor less popular items in recommendation lists significantly.

5. *Largest-Normalized-Score-First (LNSF) (Dong, Xie & Li, 2021):* this debiasing strategy seeks to achieve exposure fairness while reducing popularity bias. It uses stock volume constraints based on historical interaction frequencies to limit recommendations, thereby reducing the Matthew Effect on item popularity. It employs a minimum-cost maximum-flow model to match users and items within these constraints optimally.

6. *FA\*IR (Zehlike et al., 2017):* this method seeks to ensure a balanced representation between two groups of items in recommendations: protected and unprotected. Here, protected items correspond to long-tail items ($M \cup T$), while unprotected items represent head items ($H$). The algorithm works by maintaining separate queues for each group and merging them according to normalized scores. This approach increases the visibility of protected items, thereby mitigating the underrepresentation of long-tail items in the recommendation process.

7. *Discrepancy Minimization (DM) (Antikacioglu & Ravi, 2017):* this method focuses on increasing the aggregate diversity of recommendations, aiming to maximize the total number of unique items recommended. It leverages a minimum-cost network flow approach to identify sub-graphs that optimize diversity. The method sets a target distribution for item exposure, representing the desired frequency of each item's appearance in recommendations. The objective is to minimize the discrepancy between the actual frequency of item recommendations and this target distribution.

8. *Calibrated Popularity (CP) (Abdollahpouri et al., 2021):* this method addresses popularity bias by tailoring recommendations to a user's past interactions with items of varying popularity levels. Drawing inspiration from Steck's calibrated recommendation framework (*Steck, 2018*), CP re-ranks the initial recommendation list to align with the user's historical engagement with popular, moderately popular, and less popular items. The algorithm employs Jensen–Shannon divergence to quantify the difference between the popularity distribution in the user's profile and the recommended list, aiming to minimize this divergence for better personalization. As a user-centric approach, it uniquely focuses on adapting recommendations to individual preferences rather than applying a universal optimization. The parameter $\lambda$ regulates the trade-off between relevance and popularity calibration, ensuring that recommendations reflect user preferences across different popularity tiers. In the experiments, the candidate set size was fixed at 100, and $\lambda$ was set to 0.9, providing an optimal balance between relevance and diversity for CP.

9. *EqBal-RS (Gupta, Kaur & Jain, 2024):* this method aims to address the popularity bias prevalent in recommender systems, where popular items are disproportionately favored

over less popular ones. This approach uses a re-weighting mechanism that balances the training loss between popular and non-popular items, guided by a new metric called Popularity Parity, which measures the bias as the difference in losses. Unlike existing techniques, *EqBal-RS* eliminates the need for heavy pre-training by learning item weights dynamically during training.

## Evaluation metrics

To thoroughly assess the effectiveness of the *EquiRate* approach we introduced, we utilize a variety of standards related to the quality of recommendations in our experiments. In this process, we measure recommendation accuracy *via F1-score* and *nDCG* which are well-known and most prominent accuracy metrics. In addition, we employ four additional metrics measuring the beyond-accuracy quality of the produced recommendations. These metrics are *Average Percentage of Long Tail Items (APLT)* (*Abdollahpouri et al., 2021*; *Abdollahpouri, 2020*), *Entropy* (*Elahi et al., 2021*), *Novelty* (*Yalcin & Bilge, 2022*), and *Long-Tail Coverage (LTC)* (*Abdollahpouri, 2020*).

More importantly, these metrics evaluate recommendation lists from different perspectives. Therefore, popularity-debiasing methods can show different performances in terms of such metrics. That is, any debiasing method might show satisfactory performance for one metric, while it might not show the same performance for the other perspective of recommendation quality. Unfortunately, this makes it challenging to infer which popularity-debiasing method is the best when such different aspects of recommendation quality are concurrently considered. Therefore, as one of the main contributions of this study, we propose a novel evaluation metric, namely the *FusionIndex*, which combines these metrics in harmony. This section introduces the existing metrics and explains the proposed *FusionIndex* metric in detail.

### Existing accuracy and beyond-accuracy metrics

This section explains in detail existing accuracy (*i.e.*, *F1-score* and *nDCG*) and beyond-accuracy (*i.e.*, *APLT*, *Entropy*, *Novelty*, and *LTC*) evaluation metrics.

*F1-score*: the *F1-score* is a widely used metric to evaluate the quality of recommendations by balancing precision and recall. Specifically, it is calculated as the harmonic mean of *Precision* ($P@N_u$) and *Recall* ($R@N_u$) for a given top-*N* recommendation list for user *u*. This metric provides a single value that reflects both the accuracy of the recommended items and the coverage of relevant items, ensuring a trade-off between precision and recall. The formula for the *F1-score* is given in Eq. (4), emphasizing its role in offering a comprehensive measure of recommendation performance.

$$F1@N_u = 2 \times \frac{P@N_u \times R@N_u}{P@N_u + R@N_u}. \tag{4}$$

*Normalized Discounted Cumulative Gain (nDCG)*: the *nDCG* metric plays a significant role as a statistical measure of accuracy in our studies. It accounts for the authentic ratings given to items and their respective rankings in the top-*N* recommendations. The actual rating a user *u* gives to an item *i* is denoted as $r_{u,i}$. Utilizing this, we compute the $DCG_{N_u}^u$

and $nDCG_{N_u}^u$ for user $u$'s top-$N$ list by following the formulas outlined in Eqs. (5) and (6).

$$DCG_{N_u}^u = r_{u,i_1} + \sum_{n=2}^{N_u} \frac{r_{u,i_n}}{log_2(n)}. \tag{5}$$

$$nDCG_{N_u}^u = \frac{DCG_{N_u}^u}{IDCG_{N_u}^u}. \tag{6}$$

In this context, the $IDCG_{N_u}^u$ represents the highest potential gain for user $u$, achieved by arranging $N$ items in the most optimal sequence.

***Average Percentage of Long Tail Items (APLT):*** beyond the accuracy metrics previously detailed, we additionally apply the *Average Percentage of Long Tail Items (APLT)* to assess our ranking-based recommendations. This *APLT* index measures the representation of long-tail items within the recommended list, thus reflecting the algorithm's capacity to recommend more specialized and less common items. Consequently, this index is invaluable for evaluating the extent to which a debiasing method effectively promotes less popular (*i.e.*, tail) items in its recommendations. The identification of *tail* items within the entire catalog is informed by the Pareto principle (*Sanders, 1987*). With $T$ denoting the designated set of *tail* items, the *APLT* score for a top-$N$ list tailored to a user $u$ is determined according to the formula stated in Eq. (7).

$$APLT_u = \frac{|T \cap N_u|}{N}. \tag{7}$$

It is important to recognize that while the $APLT_u$ score demonstrates the degree of variety in recommendations for an individual user, the aggregate *APLT* score is essentially an average of these diversification levels across users. Consequently, a higher *APLT* does not always mean that the algorithm consistently delivers diverse recommendations of high quality. On the contrary, a lower *APLT* score clearly suggests a pervasive shortfall in diversity within the recommendation lists. For this reason, we also utilize *Entropy* as a supplementary measure to gauge the diversification across all recommended lists, which will be elaborated on further below.

***Entropy:*** the *Entropy* metric, as explored in *Elahi et al. (2021)*, evaluates the frequency with which different items are recommended by an algorithm across the entire catalog. Essentially, it measures the variation in how often each item is suggested. An algorithm with a higher *Entropy* score indicates a more uniform distribution of item recommendations, contributing to diversity. This uniformity is crucial for maintaining a competitive and fair market environment. To determine the *Entropy* score, we first amalgamate the top-$N$ recommendation lists from all individuals, including repetitions of items. Let $N_{all}$ be this collective set, and consider $i_1, i_2, ..., i_K$ as the complete set of catalog items, where $K$ represents the total item count. The method to calculate the *Entropy* value for a recommendation algorithm is outlined in Eq. (8).

$$Entropy = -\sum_{i \in K} (\Pr(i))log_2(\Pr(i)). \tag{8}$$

Here, $\Pr(i)$ represents the relative frequency of item $i$ appearing in the $N_{all}$ set.

*Novelty*: in our research, we use the *Novelty* metric as an additional measure that goes beyond mere accuracy. This metric assesses the capability of recommendation algorithms to suggest items that a user has not rated before. Specifically, *Novelty* quantifies the proportion of items in the top-*N* list that are new to the user, reflecting the algorithm's capacity to introduce fresh choices. To determine the *Novelty* score for a user $u$, we initially identify $I_u$ as the collection of items that user $u$ has already rated. Subsequently, the *Novelty* score for the top-*N* recommendations made to user $u$, symbolized as $Novelty_u$, is calculated according to the method detailed in Eq. (9).

$$Novelty_u = \frac{|N_u \backslash I_u|}{N}.$$ 
(9)

In this context, $N_u$ denotes the collection of the top-*N* items that have been recommended to the user $u$.

***Long-Tail Coverage (LTC)***: beyond the other metrics discussed for accuracy, our experiments also incorporate the *Long-Tail Coverage (LTC)* metric to evaluate how well recommendation algorithms represent the less popular items in the catalog, as described in *Abdollahpouri (2020)*. The *LTC* is calculated by first creating a unique aggregation of the top-*N* recommendation lists for each user, referred to as $\mathbb{N}$. In this aggregation, any repeated items in the individual top-*N* lists are removed. We then identify the shared items, $I_{\mathbb{N} \cap \mathbb{T}}$, which are present in both $\mathbb{N}$ and the tail item set $T$ as defined by the Pareto principle. The *LTC* score of a recommendation algorithm is determined by the proportion of $I_{\mathbb{N} \cap \mathbb{T}}$ relative to the total size of $T$, as detailed in Eq. (10). A higher *LTC* value suggests that the algorithm is more effective in including a broader range of less popular, or tail, items in its recommendations.

$$LTC = \frac{|I_{\mathbb{N} \cap \mathbb{T}}|}{|T|}.$$ 
(10)

### The FusionIndex: a highly useful holistic evaluation metric

In the realm of recommender systems, accurately assessing the performance of various algorithms is crucial. Traditional metrics often focus solely on accuracy or beyond-accuracy measures, failing to encapsulate the holistic performance of the system. To address this gap, we introduce the *FusionIndex*, a novel metric that harmonizes accuracy with beyond-accuracy considerations.

The *FusionIndex* is computed in two stages. First, for a given top-*N* list we form four *pairwise* harmonic means, each coupling the accuracy metric *nDCG* with one beyond-accuracy metric: $H(nDCG, APLT)$, $H(nDCG, Novelty)$, $H(nDCG, LTC)$, and $H(nDCG, Entropy)$. The harmonic mean is preferred because it moderates the influence of outliers and prevents either dimension (accuracy or diversity) from dominating the joint score. Although our experiments report both *F1-score* and *nDCG* for completeness, the latter is used inside the *FusionIndex* because it is the prevailing accuracy yardstick in recent recommender system research. Second, we take the simple arithmetic mean of these four

harmonic means to obtain a single, holistic quality indicator for the recommendation list, as formalised in Eq. (11).

$$FusionIndex = \frac{H(nDCG, APLT) + H(nDCG, Novelty) + H(nDCG, LTC) + H(nDCG, Entropy)}{4}. \quad (11)$$

This approach ensures a balanced consideration of both accuracy and beyond-accuracy aspects. By using the harmonic mean, the *FusionIndex* mitigates the risk of one aspect disproportionately influencing the overall evaluation, thereby providing a more nuanced and comprehensive assessment of recommender systems. In summary, the *FusionIndex* offers a novel and balanced methodology for evaluating recommender systems, acknowledging the importance of both accuracy and diversity in recommendations. This metric facilitates a more comprehensive understanding of the effectiveness of recommendation algorithms, paving the way for more refined and user-centric recommender systems.

## Experiment results

This section presents the findings of the experiments performed for comprehensively evaluating the proposed the *EquiRate* method and comparing it with the benchmark popularity-debiasing methods.

### In-depth analysis of the proposed EquiRate method

This section presents the improvement rates in the *FusionIndex* metric when several of the *EquiRate* variants are applied to investigate which options are the best-performing and analyze how SRI, SRG, and $\alpha$ values affect the *EquiRate* method. We employ several of the *EquiRate* variants in these experiments by considering varying SRI, SRG, and $\alpha$ values. Figures 4, 5, 6 present the best *FusionIndex* improvements by the *EquiRate* variants for the MLM, DB, and Yelp datasets, respectively. In the presented Figures, the *EquiRate* variants are labeled with the considered SRI, $\alpha$ value, and SRG methods, respectively. Note that we only employ the *FusionIndex* metric in these experiments, rather than *n*DCG, *LTC*, *Entropy*, or *APLT* solely, as it evaluates the final recommendations by considering both accuracy and beyond-accuracy aspects appropriately and concurrently, as discussed previously.

As can be followed by Fig. 4, in the MLM dataset, the *HIFA (0.9)_GRV-$i_{avg}$* variant of the *EquiRate* achieved the most effective result with a 31.29% improvement in the *FusionIndex* metric. This outcome demonstrates the effectiveness of the HIFA strategy and the high $\alpha$ value in reducing the popularity bias. This strategy has particularly balanced the overrepresentation of popular products, thereby increasing the visibility of lesser-known content. The *TPA (0.9)_GRV-$i_{avg}$* variant, with its 26.20% improvement, ranks second, indicating the efficacy of the threshold-based approach in minimizing the gap between popular and less popular items. The high $\alpha$ value proves to be an effective tool in correcting popularity imbalance. The *HIFA (0.5)_GRV-$i_{avg}$* variant, with a 21.65% improvement, offers a more balanced enhancement with a lower $\alpha$ value, playing a significant role in improving both the accuracy and diversity of recommender systems.

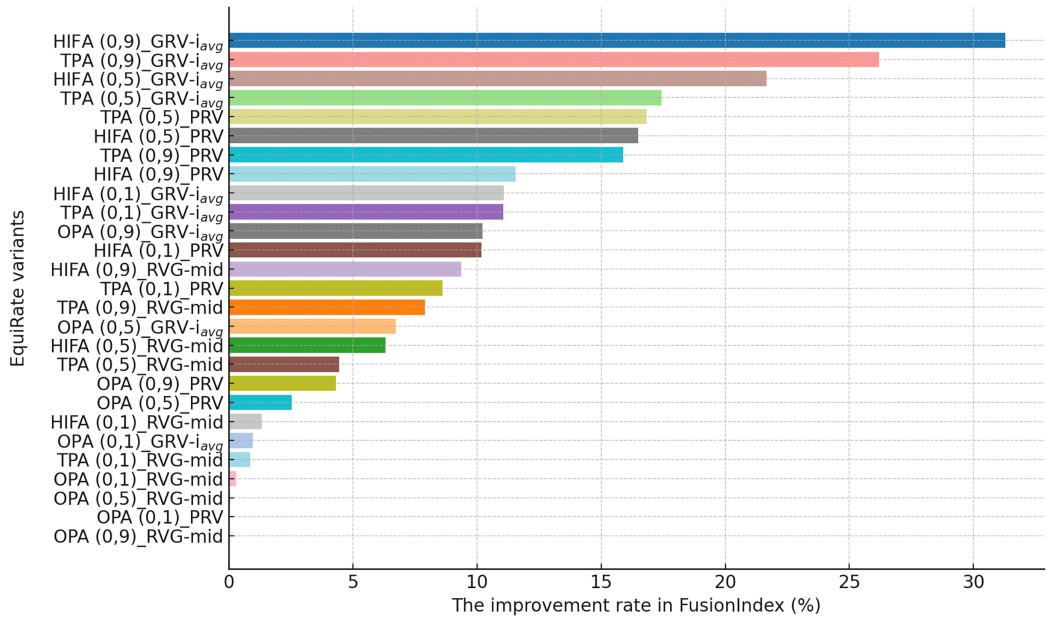

**Figure 4** **The improvement rate in the *FusionIndex* (%) with several variants of the *EquiRate* method in the MLM dataset.** Here, the *EquiRate* variants are labeled with the considered SRI, $\alpha$ value, and SRG methods, respectively.

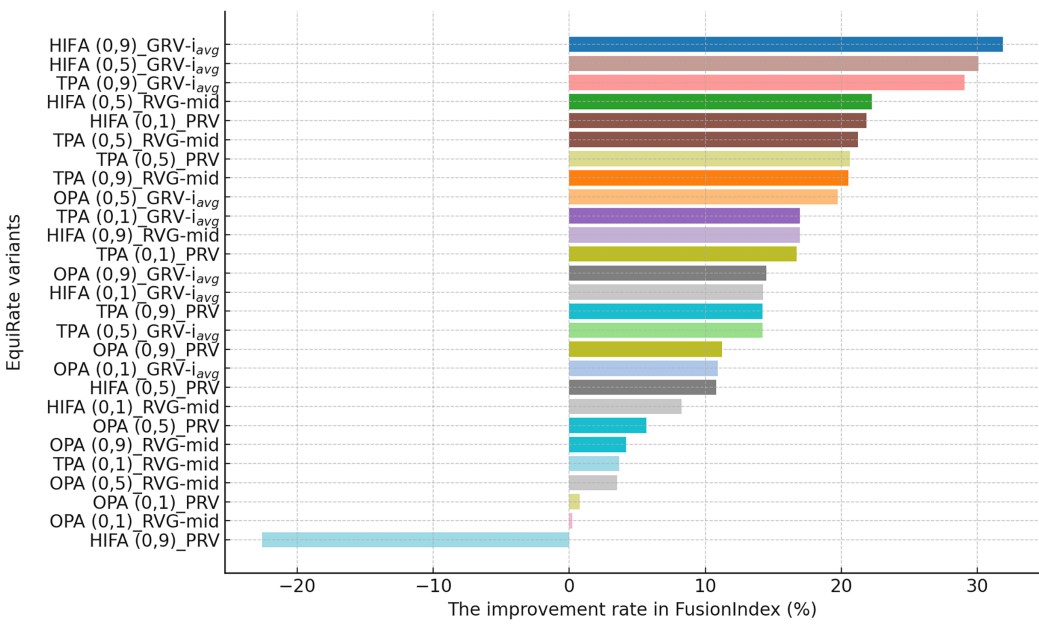

**Figure 5** **The improvement rate in the *FusionIndex* (%) with several variants of the *EquiRate* method in the DB dataset.** Here, the *EquiRate* variants are labeled with the considered SRI, $\alpha$ value, and SRG methods, respectively.

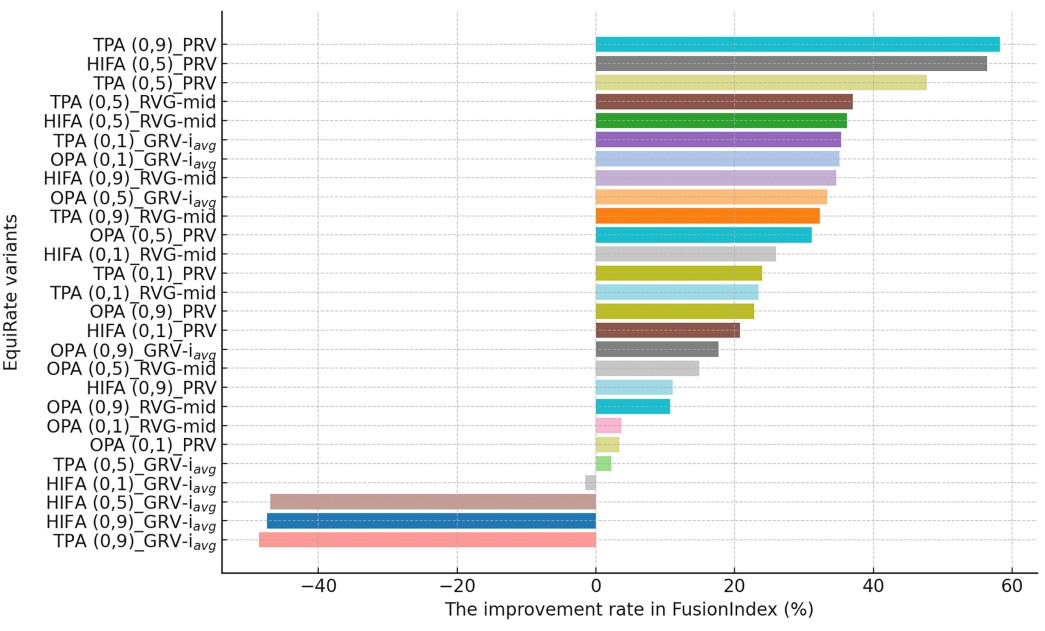

**Figure 6 The improvement rate in the *FusionIndex* (%) with several variants of the *EquiRate* method in the Yelp dataset.** Here, the *EquiRate* variants are labeled with the considered SRI, $\alpha$ value, and SRG methods, respectively.

On the other hand, for the DB dataset, the *HIFA (0.9)_GRV-$i_{avg}$* variant's success is particularly notable, as can be followed by Fig. 5. In environments with sparse data structures, this strategy shows how high $\alpha$ value can produce effective results. It enhances the visibility of less popular content in recommendation lists by injecting synthetic ratings. The *HIFA (0.5)_GRV-$i_{avg}$* and *TPA (0.9)_GRV-$i_{avg}$* variants also yield effective outcomes with medium and high $\alpha$ values, successfully reducing the popularity bias in the DB dataset. Note also that these results are highly parallel with those observed for the MLM dataset.

Finally, according to the presented results for the Yelp dataset in Fig. 6, the *TPA (0.9) _PRV* variant achieving the highest improvement at 58.28% underscores the strategy's effectiveness in large and sparse data structures. The integration of the PRV method with a personalized recommender system provides an effective solution in such complex data structures. The *HIFA (0.5)_PRV* and *TPA (0.5)_PRV* variants also demonstrate significant improvements with medium $\alpha$ values, showing their efficacy in reducing popularity bias and enhancing the diversity of recommendations in the Yelp dataset. Also, we observe that the best-performing the *EquiRate* variants in Yelp are slightly different from those in both MLM and DB, indicating that the *EquiRate* variants can show different performances depending on the utilized dataset. Nevertheless, we can conclude that significant improvements in the *FusionIndex* metric can be achieved with a proper parameter-tuning of the *EquiRate* in any dataset.

The performance analysis of the *EquiRate* method in the MLM, DB, and Yelp datasets illustrates its ability to reduce popularity bias in recommender systems and to enhance the

diversity of recommendation lists. The results observed in each dataset highlight the application of the *EquiRate*'s SRI and SRG strategies and particularly the role of the $\alpha$ scaling factor. More specifically, in the MLM dataset, the combination of high $\alpha$ values with HIFA and TPA strategies offers an aggressive approach to correcting popularity imbalance. This significantly increased the visibility of lesser-known or less-evaluated items in recommendation lists, enhancing user satisfaction and system effectiveness. In the DB dataset, the high $\alpha$ value combined with the HIFA strategy proved to be effective in correcting popularity imbalance in sparse data structures. In the Yelp dataset, the combination of the TPA strategy and the PRV method in large and complex data structures significantly enhanced both the accuracy and diversity of the recommendation system.

These outcomes demonstrate how the *EquiRate*'s SRI methods (OPA, HIFA, and TPA) and SRG methods (GRV, PRV, and RVG) produce different results at various $\alpha$ values. In particular, high $\alpha$ values provide a more aggressive approach to correcting popularity imbalance, while medium $\alpha$ values offer more balanced outcomes. Personalization methods like PRV enhance the accuracy of recommendation systems by reflecting user preferences more effectively, whereas methods like GRV and RVG increase the visibility of diversity and undiscovered content.

Also, the reliability of the experimental results is strengthened by conducting multiple iterations and taking the average of these repetitions. This approach ensures the stability and reliability of the outcomes, providing a more accurate reflection of the method's effectiveness. In a nutshell, the improvements achieved by the various variants of the *EquiRate* method in reducing popularity bias and enhancing the diversity and accuracy of recommender systems across different datasets are significant. The method's adaptability to the unique characteristics of each dataset and the importance of the $\alpha$ scaling factor in achieving these improvements are evident. This highlights the potential of the *EquiRate* in enhancing recommender systems and enriching the user experience.

### Comparison of the EquiRate with existing popularity-debiasing methods

In this section, we present the findings of additional experiments where we compared best-performing the *EquiRate* variants with several benchmark popularity-debiasing methods. These benchmarks are *AUG*, *xQuad*, *PAW*, *MUL*, *LNSF*, *CP*, *DM*, *FA\*IR*, and *EqBal-RS*, whose details are explained previously. In these experiments, we consider the best ten variants of the *EquiRate* according to each dataset, which are discovered in the experiments in the previous section.

Although the proposed *FusionIndex* is an advanced metric considering accuracy and beyond-accuracy concurrently, unlike the previous experiments, we present both the *FusionIndex* and each accuracy (*i.e.*, *F1-score* and *n*DCG) and beyond-accuracy (*i.e.*, *APLT*, *Novelty*, *LTC*, and *Entropy*) outcomes in these experiments to provide a better picture. Accordingly, we first present the accuracy and beyond-accuracy results of the recommendation lists produced *via* our *EquiRate* variants and other existing benchmarks for the MLM, DB, and Yelp datasets in Tables 5, 6, 7, respectively. Note that we also present the results of the original recommendation algorithm in these tables before any

**Table 5** Comparison of *EquiRate* variants with other popularity-debiasing methods on the MLM dataset for *F1-score*, *n* DCG, *APLT*, *Novelty*, *LTC*, and *Entropy*.

| Debiasing method | SRI | α | SRG | Approach | F1-score | nDCG | APLT | Novelty | LTC | Entropy |
|---|---|---|---|---|---|---|---|---|---|---|
| *Original* | | | | | **0.161** | **0.620** | 0.295 | 0.296 | 0.150 | 0.636 |
| *EquiRate Variants* | HIFA | 0.9 | GRV | $i_{avg}$ | 0.116 | 0.556 | 0.417 | 0.367 | 0.744 | 0.747 |
| | TPA | 0.9 | GRV | $i_{avg}$ | 0.135 | 0.597 | 0.383 | 0.323 | 0.550 | 0.714 |
| | HIFA | 0.5 | GRV | $i_{avg}$ | 0.141 | 0.608 | 0.366 | 0.311 | 0.445 | 0.699 |
| | TPA | 0.5 | GRV | $i_{avg}$ | 0.143 | 0.610 | 0.367 | 0.310 | 0.350 | 0.689 |
| | TPA | 0.5 | PRV | | 0.115 | 0.534 | 0.413 | 0.400 | 0.350 | 0.702 |
| | HIFA | 0.5 | PRV | | 0.101 | 0.492 | 0.434 | 0.447 | 0.388 | 0.715 |
| | TPA | 0.9 | PRV | | 0.090 | 0.459 | 0.483 | 0.484 | 0.407 | 0.728 |
| | HIFA | 0.9 | PRV | | 0.074 | 0.401 | 0.536 | 0.548 | 0.443 | 0.743 |
| | HIFA | 0.1 | GRV | $i_{avg}$ | 0.138 | 0.597 | 0.364 | 0.327 | 0.246 | 0.671 |
| | TPA | 0.1 | GRV | $i_{avg}$ | 0.138 | 0.595 | 0.368 | 0.329 | 0.242 | 0.674 |
| *AUG* | | | | | 0.102 | 0.452 | 0.827 | 0.453 | 0.285 | 0.783 |
| *xQuad* | | | | | 0.145 | 0.568 | 0.630 | 0.338 | 0.171 | 0.691 |
| *PAW* | | | | | 0.161 | 0.620 | 0.302 | 0.296 | 0.158 | 0.639 |
| *MUL* | | | | | 0.056 | 0.309 | **0.994***| 0.607 | 0.396 | 0.812 |
| *LNSF* | | | | | 0.003 | 0.356 | 0.831 | **0.984***| **0.936***| **0.916***|
| *CP* | | | | | 0.106 | 0.434 | 0.179 | 0.537 | 0.165 | 0.369 |
| *DM* | | | | | 0.013 | 0.393 | 0.913 | 0.896 | 0.186 | 0.372 |
| *FA*IR* | | | | | 0.087 | 0.403 | 0.124 | 0.572 | 0.215 | 0.332 |
| *EqBal-RS* | | | | | 0.089 | 0.483 | 0.614 | 0.602 | 0.195 | 0.352 |

**Note:**
Bold numbers indicate the best score per metric; the symbol '*' marks a result that is significantly better than the second-best at the 99% confidence level.

**Table 6** Comparison of *EquiRate* variants with other popularity-debiasing methods on the DB dataset for *F1-score*, *nDCG*, *APLT*, *Novelty*, *LTC*, and *Entropy*.

| Debiasing Method | SRI | α | SRG | Approach | F1-score | nDCG | APLT | Novelty | LTC | Entropy |
|---|---|---|---|---|---|---|---|---|---|---|
| *Original* | | | | | **0.097** | **0.283** | 0.079 | 0.715 | 0.016 | 0.457 |
| *EquiRate Variants* | HIFA | 0.9 | GRV | $i_{avg}$ | 0.045 | 0.189 | 0.664 | 0.796 | **0.658***| **0.808***|
| | HIFA | 0.5 | GRV | $i_{avg}$ | 0.056 | 0.213 | 0.613 | 0.771 | 0.216 | 0.637 |
| | TPA | 0.9 | GRV | $i_{avg}$ | 0.056 | 0.216 | 0.588 | 0.769 | 0.190 | 0.612 |
| | HIFA | 0.5 | RVG | *mid* | 0.083 | 0.259 | 0.477 | 0.729 | 0.025 | 0.528 |
| | HIFA | 0.1 | PRV | | 0.092 | 0.271 | 0.344 | 0.730 | 0.026 | 0.545 |
| | TPA | 0.5 | RVG | *mid* | 0.094 | 0.279 | 0.336 | 0.720 | 0.018 | 0.511 |
| | TPA | 0.5 | PRV | | 0.073 | 0.229 | 0.534 | 0.761 | 0.059 | 0.603 |
| | TPA | 0.9 | RVG | *mid* | 0.082 | 0.255 | 0.476 | 0.734 | 0.023 | 0.523 |
| | OPA | 0.5 | GRV | $i_{avg}$ | 0.079 | 0.251 | 0.490 | 0.737 | 0.023 | 0.529 |
| | TPA | 0.1 | GRV | $i_{avg}$ | 0.067 | 0.230 | 0.560 | 0.754 | 0.035 | 0.539 |
| *xQuad* | | | | | 0.081 | 0.227 | 0.581 | 0.767 | 0.027 | 0.515 |
| *AUG* | | | | | 0.077 | 0.224 | 0.454 | 0.771 | 0.039 | 0.581 |
| *PAW* | | | | | 0.097 | 0.280 | 0.095 | 0.718 | 0.019 | 0.465 |

(Continued)

| Debiasing Method | SRI | α | SRG | Approach | F1-score | nDCG | APLT | Novelty | LTC | Entropy |
|---|---|---|---|---|---|---|---|---|---|---|
| MUL | | | | | 0.042 | 0.139 | 0.791 | 0.846 | 0.060 | 0.606 |
| LNSF | | | | | 0.001 | 0.238 | 0.774 | 0.793 | 0.048 | 0.573 |
| CP | | | | | 0.058 | 0.212 | 0.454 | 0.847 | 0.036 | 0.434 |
| DM | | | | | 0.001 | 0.202 | **1.000**\* | 0.995 | 0.086 | 0.351 |
| FA\*IR | | | | | 0.002 | 0.195 | **1.000**\* | **0.996** | 0.094 | 0.493 |
| EqBal-RS | | | | | 0.072 | 0.235 | 0.855 | 0.947 | 0.087 | 0.475 |

Note:
Bold numbers indicate the best score per metric; the symbol '*' marks a result that is significantly better than the second-best at the 99% confidence level.

**Table 7 Comparison of *EquiRate* variants with other popularity-debiasing methods on the Yelp dataset for *F1-score*, *nDCG*, *APLT*, *Novelty*, *LTC*, and *Entropy*.**

| Debiasing Method | SRI | α | SRG | Approach | F1-score | nDCG | APLT | Novelty | LTC | Entropy |
|---|---|---|---|---|---|---|---|---|---|---|
| Original | | | | | 0.0214 | 0.046 | 0.072 | 0.964 | 0.014 | 0.255 |
| EquiRate Variants | TPA | 0.9 | PRV | | 0.0213 | 0.057 | 0.694 | 0.964 | **0.118**\* | 0.275 |
| | HIFA | 0.5 | PRV | | 0.0175 | 0.057 | 0.709 | 0.968 | 0.105 | 0.266 |
| | TPA | 0.5 | PRV | | 0.0206 | 0.056 | 0.443 | 0.964 | 0.063 | 0.255 |
| | TPA | 0.5 | RVG | $mid$ | 0.0200 | 0.059 | 0.360 | 0.964 | 0.014 | 0.265 |
| | HIFA | 0.5 | RVG | $mid$ | 0.0180 | 0.056 | 0.511 | 0.967 | 0.021 | 0.259 |
| | TPA | 0.1 | GRV | $i_{avg}$ | 0.0157 | 0.054 | 0.448 | 0.970 | 0.028 | 0.266 |
| | OPA | 0.1 | GRV | $i_{avg}$ | 0.0208 | **0.061**\* | 0.156 | 0.963 | 0.015 | 0.262 |
| | HIFA | 0.9 | RVG | $mid$ | 0.0166 | 0.053 | 0.533 | 0.970 | 0.034 | 0.255 |
| | OPA | 0.5 | GRV | $i_{avg}$ | 0.0191 | 0.056 | 0.476 | 0.966 | 0.018 | 0.250 |
| | TPA | 0.9 | RVG | $mid$ | 0.0209 | 0.055 | 0.429 | 0.965 | 0.019 | 0.252 |
| xQuad | | | | | 0.0166 | 0.047 | 0.496 | 0.972 | 0.002 | 0.286 |
| LNSF | | | | | 0.0094 | 0.051 | 0.562 | 0.979 | 0.000 | 0.277 |
| PAW | | | | | 0.0207 | 0.055 | 0.000 | 0.964 | 0.000 | 0.256 |
| AUG | | | | | 0.0122 | 0.028 | 0.050 | 0.975 | 0.001 | 0.262 |
| MUL | | | | | 0.0042 | 0.010 | 0.797 | 0.990 | 0.001 | 0.261 |
| CP | | | | | **0.0329**\* | 0.048 | 0.221 | 0.963 | 0.009 | **0.438**\* |
| DM | | | | | 0.0013 | 0.031 | **0.915**\* | **0.997** | 0.002 | 0.290 |
| FA\*IR | | | | | 0.0148 | 0.036 | 0.256 | 0.979 | 0.004 | 0.403 |
| EqBal-RS | | | | | 0.0160 | 0.049 | 0.876 | 0.987 | 0.008 | 0.251 |

Note:
Bold numbers indicate the best score per metric; the symbol '*' marks a result that is significantly better than the second-best at the 99% confidence level.

popularity-debiasing method is applied; thus, we can conclude how much the popularity-debiasing method can mitigate this issue in the final recommendations. In each results table, the top-performing method for a given metric is shown in bold. Additionally, the symbol '*' appended to a bold value indicates that its improvement over the second-best method is statistically significant at the 99% confidence level. To provide an overall evaluation, we also present the improvement ratio in the *FusionIndex via* the best-performing ten *EquiRate* variants and existing popularity debiasing methods for the MLM, DB and Yelp data sets in Figs. 7, 8, 9, respectively.
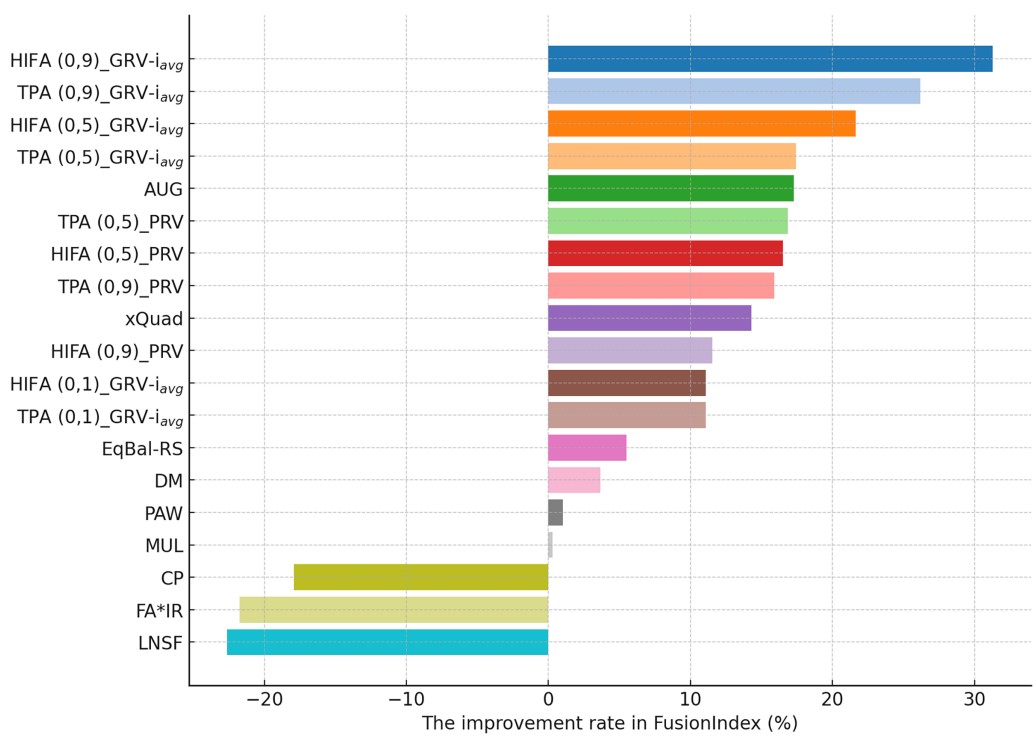

**Figure 7 The improvement rates in the *FusionIndex* when popularity-debiasing methods are applied for the MLM dataset.**

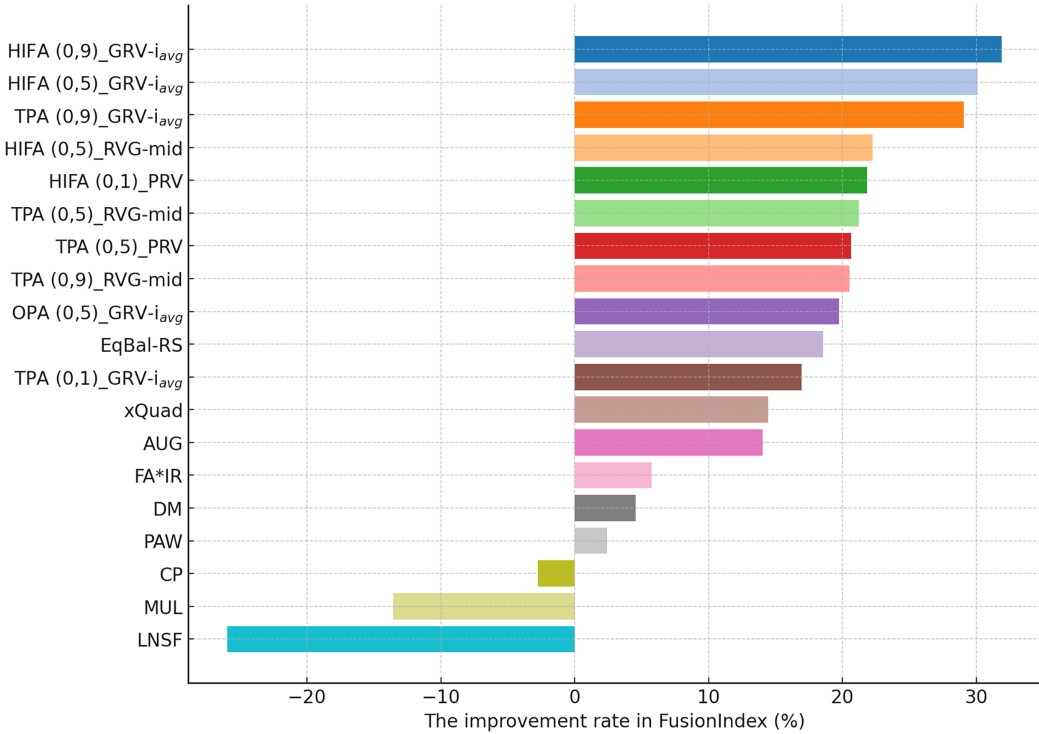

**Figure 8 The improvement rates in the *FusionIndex* when popularity-debiasing methods are applied for the DB dataset.**

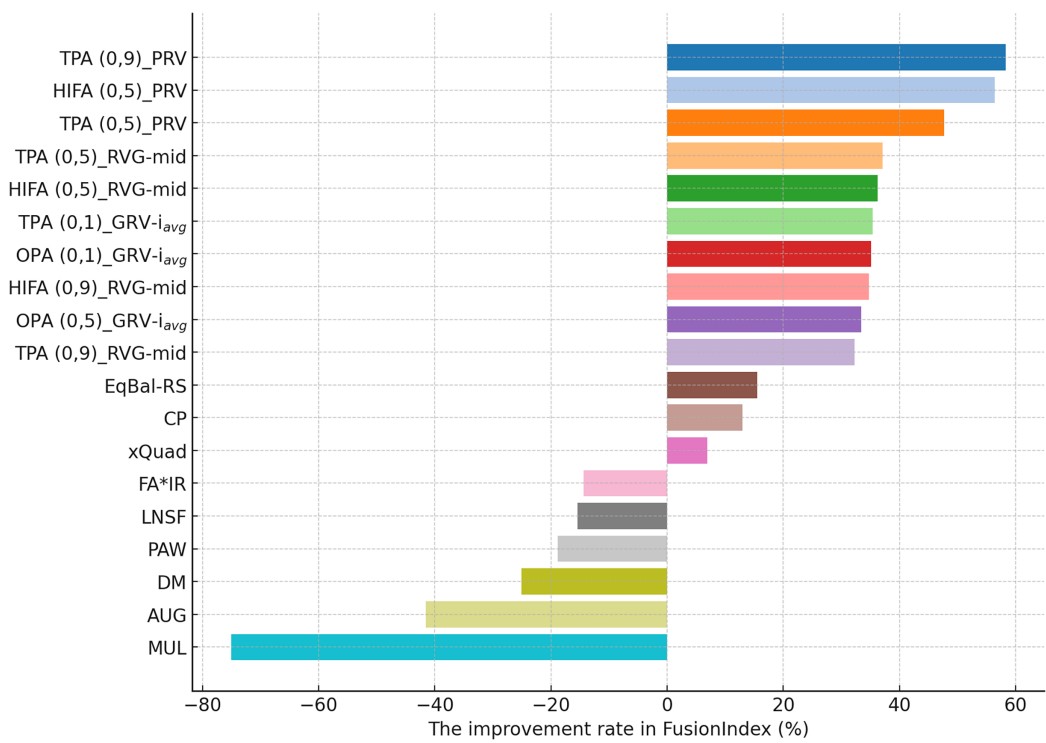

**Figure 9** **The improvement rates in the *FusionIndex* when popularity-debiasing methods are applied for the Yelp dataset.**

As shown in Table 5, the original algorithm achieves an *n*DCG of 0.620 on the MLM dataset before applying any debiasing methods. Unfortunately, all *EquiRate* variants yield slightly lower *n*DCG values. Similarly, the *F1-scores* of all *EquiRate* variants are also lower than those of the original algorithm. This suggests a minor reduction in accuracy, which aligns with expectations given the trade-off between accuracy and beyond-accuracy objectives in mitigating popularity bias. Nonetheless, the relatively high *F1-score* and *n*DCG indicate that the recommended items remain relevant to users. Furthermore, compared to other debiasing methods, particularly *MUL*, which tend to cause greater declines in *n*DCG, the performance drop with *EquiRate* variants is more acceptable, reinforcing the method's balance between accuracy and fairness.

In *APLT*, with an original value of 0.295, significant increases are observed with high $\alpha$ values in the *EquiRate* (*e.g.*, 0.536 in case SRI is as HIFA, $\alpha = 0.9$, and SRG is as PRV), as shown in Table 5. This indicates a tendency of recommender systems, enhanced with our *EquiRate* method, to suggest less popular, long-tail items, thus providing more diverse content for users. Similarly, for *Novelty*, originally at 0.296, various the *EquiRate* variants have increased this value. Specifically, the same *EquiRate* variant above, with a 0.548 *Novelty* score, has significantly enhanced the level of *Novelty* in the recommendation lists. This reflects a trend towards recommending more innovative items. In *LTC*, originally at 0.150, high $\alpha$ value the *EquiRate* variants have shown significant increases (*e.g.*, 0.744 in case SRI is as HIFA, $\alpha = 0.9$, and SRG is as GRV-$i_{avg}$), indicating that more long-tail items

are covered in recommendation lists, thereby increasing the visibility of less popular items. Lastly, *Entropy*, originally at 0.636, has generally increased with the *EquiRate* variants. High $\alpha$ value variants (*e.g.*, 0.747 in case the variant above is used) exhibit higher *Entropy* values. This demonstrates that recommendation lists have become more diverse, offering users a broader range of items. In conclusion, various *EquiRate* variants, particularly those with high $\alpha$ values, effectively reduce popularity bias and enhance diversity in the MLM dataset. These results suggest a potential to improve the balance between accuracy and diversity.

While some existing popularity-debiasing methods (*e.g.*, *MUL*, *LNSF*, and DM) may outperform the *EquiRate* variants in beyond-accuracy metrics, they come at the cost of significantly reducing *F1-score* and *n*DCG, as shown in Table 5. This is undesirable since the primary goal of mitigating popularity bias is to improve beyond-accuracy performance without sacrificing too much accuracy. Evaluating each metric separately, however, can make it difficult to determine which debiasing method offers the best overall performance. To address this, we present the improvement ratios in the *FusionIndex* in Fig. 7, offering a more comprehensive view of the overall effectiveness of various *EquiRate* variants and other debiasing methods on the MLM dataset.

As illustrated in Fig. 7, the *EquiRate* variants, particularly those with high $\alpha$ values such as HIFA and TPA, achieve the highest *FusionIndex* scores. Notably, the HIFA (0.9)_GRV-$i_{avg}$ variant records the top score of 31.286%, followed closely by the TPA (0.9)_GRV-$i_{avg}$ variant with 26.203%. These results highlight the effectiveness of these variants in improving diversity and novelty in recommendation lists. Moreover, they emphasize the crucial role of high $\alpha$ values in mitigating popularity bias and enhancing diversity. While other debiasing methods, except for *CP*, *FA*$^*$*IR*, and *LNSF*, show positive *FusionIndex* outcomes, the *EquiRate* variants consistently outperform these benchmarks on the MLM dataset.

As shown in Table 6, the results for the DB dataset reveal that all *EquiRate* variants result in lower *F1-score* and *n*DCG values compared to the original scores of 0.097 and 0.283, respectively, similar to the findings for the MLM dataset. However, strategies with high $\alpha$ values, such as TPA (*e.g.*, *n*DCG of 0.279 when SRI is TPA with $\alpha = 0.5$ and SRG is RVG$_{mid}$) and HIFA (*e.g.*, *n*DCG of 0.271 when SRI is HIFA with $\alpha = 0.1$ and SRG is PRV), result in only slight decreases in *n*DCG, maintaining reasonable accuracy. This trend is consistent for *F1-score* as well. In contrast, other popularity-debiasing methods tend to cause more substantial declines in both *n*DCG (*e.g.*, *MUL*) and *F1-score* (*e.g.*, *DM*), similar to the MLM dataset results.

In *APLT*, originally valued at 0.079, significant increases have been observed with high $\alpha$ values in the *EquiRate* (*e.g.*, 0.664 in case SRI is as HIFA, $\alpha = 0.9$, and SRG is as GRV-$i_{avg}$). This implies a tendency of the recommender system to suggest less popular, long-tail items, thus including more diverse content in the recommendations. Similarly, for *Novelty*, originally at 0.715, various *EquiRate* variants have increased this value. Particularly, the variant mentioned above reaches the maximum level of *Novelty* in the recommendation lists, *i.e.*, 0.796, suggesting a shift towards recommending more innovative items. Originally at 0.016, *LTC* has shown significant increases with high $\alpha$ value *EquiRate* variants (*e.g.*, 0.658

in case the same variant). This indicates broader coverage of long-tail items in the recommendation lists, enhancing the visibility of less popular items. Originally at 0.457, *Entropy* has generally increased with the *EquiRate* variants. Specifically, the same variant achieves the highest *Entropy* values (*i.e.*, 0.808), indicating more diverse recommendation lists and offering users a wider range of items. In conclusion, the *EquiRate* and its variants effectively reduce popularity bias and enhance diversity in the DB dataset.

Unlike the MLM dataset, the DB dataset shows that the best-performing *EquiRate* variant significantly outperforms other debiasing strategies in terms of *n*DCG, *Entropy*, and *LTC*, as shown in Table 6. For other metrics, such as *F1-score*, *APLT*, and *Novelty*, the top *EquiRate* variant delivers results comparable to those of benchmark methods.

The improvement rates in the *FusionIndex*, illustrated in Fig. 8, show that all *EquiRate* variants outperform the benchmark popularity-debiasing methods on the DB dataset. Specifically, HIFA variants (HIFA (0.9)_GRV-$i_{avg}$, HIFA (0.5)_GRV-$i_{avg}$) and TPA (0.9)_GRV-$i_{avg}$ achieve the highest *FusionIndex* scores, with 31.906%, 30.085%, and 29.068%, respectively. These results highlight the effectiveness of *EquiRate* in mitigating popularity bias and enhancing diversity in recommendation lists. Interestingly, the *LNSF* and *MUL* methods lead to a significant drop in *FusionIndex* scores, primarily due to their adverse impact on overall accuracy, which negatively affects the *FusionIndex*. A similar trend for *LNSF* is observed in the MLM dataset, as shown in Fig. 7.

When considering the accuracy and beyond-accuracy outcomes for the Yelp dataset from Table 7, for *n*DCG, with an original value of 0.046, various *EquiRate* variants have generally increased this metric. This observation differs from the ML and DB datasets, where our *EquiRate* variants led to negligible decreases in *n*DCG values, as previously discussed. However, this observation is not valid for other accuracy metrics, *i.e.*, *F1-score*. Due to the trade-off between accuracy and beyond-accuracy aspects of the recommendations, improvements in accuracy *via* the *EquiRate* variants unfortunately do not translate into significant improvements in *Entropy* and *Novelty*, as shown in Table 7. Nevertheless, we can conclude that all the *EquiRate* variants significantly enhance the *APLT* of the recommendations, and some (*e.g.*, TPA (0.9)_PRV) provide notable improvements in the *LTC*. In conclusion, our *EquiRate* variants remarkably enhance the likelihood of including tail items in the final recommendations by simultaneously treating popularity bias and providing more accurate recommendations than the original version of the recommendation algorithm.

Table 7 demonstrates that the top-performing *EquiRate* variant significantly outperforms other debiasing methods in *n*DCG and *LTC* metrics. For other metrics, it delivers results comparable to those of the benchmark approaches. Additionally, as shown in Fig. 9, the *EquiRate* methods—particularly those using the TPA strategy—achieve the highest *FusionIndex* scores on the Yelp dataset. Notably, the improvement in *FusionIndex* for the Yelp dataset is much greater than that observed for the MLM and DB datasets. This highlights the practical applicability of *EquiRate*, as the Yelp dataset, being sparser and larger, provides a more realistic test environment, closely resembling real-world scenarios.

As depicted in Fig. 9, all benchmark debiasing methods generally achieve lower *FusionIndex* scores on the Yelp dataset compared to the *EquiRate* variants. Notably, most

methods—except for *EqBal-RS*, *CP*, and *xQuad*—fail to improve *FusionIndex* scores, with some, like *MUL*, even causing significant declines. While traditional popularity-debiasing methods are designed to address specific aspects of recommendations by reducing popularity bias, our analysis using the proposed *FusionIndex* metric highlights their limitations in simultaneously balancing accuracy and multiple well-known beyond-accuracy metrics.

## INSIGHTS AND DISCUSSION

According to the performed extensive set of experiments based on three famous datasets, we give the most critical gained insights in the following.

- The *EquiRate* variants that select HIFA or TPA as the SRI strategy, as opposed to OPA, can more effectively mitigate popularity bias and achieve higher-quality recommendations. The main reason for this observation is that both HIFA and TPA strategies consider only head items in the catalog when calculating the average popularity score, which is later used to determine the total number of ratings to be injected. In contrast, the OPA strategy considers the average popularity of all items in the catalog when calculating the number of ratings to inject. This results in more ratings being injected into the profiles of tail items with HIFA and TPA than with OPA, thereby more effectively addressing the issue of popularity bias with these two strategies.

- Our proposed *EquiRate* method can more effectively address the popularity bias problem and achieve higher-quality recommendations, particularly in terms of beyond-accuracy aspects, when designed with higher $\alpha$ values. This effectiveness stems from the fact that higher $\alpha$ values entail injecting more synthetic ratings into the profiles of selected items. This approach helps to reduce imbalances in the rating distribution within the data, which is the primary cause of the popularity bias issue in recommendation algorithms.

- In comparing SRG strategies, our *EquiRate* variants typically exhibit better performance when the GRV strategy is implemented. This approach, which involves averaging the ratings of items (*i.e.*, $i_{avg}$), proves more effective than the other two methods. The advantage of the GRV strategy lies in its ability to preserve the original rating patterns of item profiles without distorting them through the injection of average ratings.

- Our experiments demonstrate that traditional popularity-debiasing methods exhibit varied performances across different beyond-accuracy evaluation criteria, even while addressing the issue of popularity bias. This variation primarily stems from the fact that these methods are tailored to enhance specific aspects of recommendations, complicating the process of forming a comprehensive evaluation. In contrast, our proposed *FusionIndex* metric offers a well-balanced and uniform criterion for evaluating both the accuracy and beyond-accuracy aspects of recommendations. The experiments conducted using the *FusionIndex* demonstrate that existing popularity-debiasing methods can sometimes be ineffective since they fail to improve the *FusionIndex* scores of the recommendations and may even result in decreases.

 

- The best-performing *EquiRate* variants typically surpass other debiasing strategies, even though they may occasionally show poorer performance in certain specific evaluation criteria. Conversely, they tend to yield more significant improvements in the *FusionIndex* score of the recommendations compared to existing strategies. This observation holds across all datasets and is particularly pronounced for Yelp. It's important to note that the Yelp dataset, with its high dimensionality and sparsity ratio, is more representative of real-world scenarios, demonstrating the effectiveness of our *EquiRate* method in providing more diverse and fair recommendations more clearly.

The extensive research conducted on three well-known datasets reveals that the *EquiRate* approach, especially through its HIFA and TPA strategies, effectively reduces popularity bias in recommendation systems. These strategies surpass the OPA strategy by strategically utilizing head item data to adjust recommendations. Higher $\alpha$ values are crucial in enhancing the quality of recommendations, particularly for metrics beyond accuracy. The GRV strategy is noted for preserving genuine rating patterns, avoiding artificial distortion. This research also exposes the shortcomings of traditional debiasing methods and emphasizes the comprehensive evaluation capabilities of the *FusionIndex*. Remarkably, the *EquiRate* demonstrates strong performance across all datasets, with its success most apparent in the Yelp dataset, highlighting its potential in real-world applications and contributing to the advancement of fair and diverse recommender systems.

In conclusion, the *EquiRate* method offers significant potential for real-world application, particularly in industrial platforms such as e-commerce, streaming services, and online marketplaces. By addressing the root cause of popularity bias through synthetic rating injection, our method ensures a more balanced exposure of items in recommendation lists. This balance not only promotes fairness but also creates opportunities for smaller vendors or content creators to compete with established, popular items. Such an approach can lead to a more diverse and equitable distribution of sales and content consumption, fostering a healthier ecosystem for both providers and users.

From a user perspective, the reduction in popularity bias enhances the discovery of personalized and niche content, improving the overall user experience. This increased satisfaction can translate into longer engagement periods and greater loyalty to the platform. Additionally, industrial platforms can benefit from higher customer retention rates as users feel their preferences are better understood and catered to. These factors highlight the practicality and value of adopting the *EquiRate* method for platforms aiming to balance commercial success with user satisfaction and fairness.

## CONCLUSION AND FUTURE WORK

Popularity bias in recommender systems predominantly favors well-known items, often at the expense of lesser-known or niche ones. This can stifle recommendation diversity and limit the discovery of a broader range of content, impacting the user experience. Addressing this bias is essential to ensure a more balanced and inclusive item exposure, which in turn can increase user satisfaction by revealing undiscovered content and creating an equitable platform for all content creators.

The *EquiRate* popularity-debiasing method, as proposed in this study, offers a recalibrated recommendation approach that injects synthetic ratings into less popular, or "tail," items to ensure a balanced representation. This allows items that are typically less visible to have a fair chance of being recommended. The proposed *EquiRate* first classifies items using the famous Pareto principle and then applies three *Synthetic Rating Injection (SRI)* strategies; *Overall Popularity Adjustment (OPA)*, *Head Item Focused Adjustment (HIFA)*, and *Threshold-Based Popularity Adjustment (TPA)*, for strategic allocation and computation of synthetic ratings. Additionally, three Synthetic *Rating Generation (SRG)* strategies; *Generalized Rating Value (GRV)*, *Predictive Rating Value (PRV)*, and *Randomized Value Generation (RVG)*, are used to uphold the authentic profile of items while diminishing popularity bias. The nine variants of the *EquiRate*, derived from combinations of *SRI* and *SRG* strategies, undergo experimental validation to identify the most effective one for debiasing. The selected variant then preprocesses data for recommendation algorithms, creating a balanced dataset that betters the representation of less popular items and enhances the diversity, novelty, and fairness of recommendations, promoting a more equitable environment. Additionally, the *FusionIndex* is introduced as an advanced metric that evaluates recommendation lists by concurrently measuring both accuracy and a spectrum of beyond-accuracy aspects such as diversity, catalog coverage, novelty, and fairness. This holistic metric is particularly valuable for assessing the efficacy of strategies designed to mitigate popularity bias in recommender systems.

The experiment results for the well-known MovieLens-1M (MLM), DoubanBook (DB), and Yelp datasets provide insightful evaluations of the *EquiRate* method compared to benchmark popularity-debiasing methods. Across all three datasets, several *EquiRate* variants significantly outperform the five prominent popularity-debiasing methods in terms of the *FusionIndex* metric. More specifically, in both MLM and DB datasets, the *HIFA-GRV* variant of the *EquiRate* usually achieves the highest improvement, effectively balancing the overrepresentation of popular items. On the other hand, for Yelp, the *TPA-PRV* variant stood out, indicating its effectiveness in large and sparse datasets. These results highlight the *EquiRate*'s efficacy in reducing popularity bias and enhancing the diversity of recommendations. Additionally, although some existing methods for reducing popularity bias in recommender systems show promising results in specific beyond-accuracy metrics, they generally attain lower overall *FusionIndex* scores compared to our top-performing the *EquiRate* variants. Notably, when evaluated using the *FusionIndex*, these methods often prove significantly less effective in balancing accuracy with beyond-accuracy factors, failing to enhance the *FusionIndex* scores of the recommendations and in some cases, even causing substantial declines.

We also suggest three future research directions for this study: firstly, exploring different algorithms for the *Predictive Rating Value (PRV)* within Synthetic Rating Injection methods, where currently the VAECF algorithm is employed. Secondly, the paragraph proposes enriching and enhancing the *FusionIndex* metric by incorporating additional metrics such as the average popularity of the recommended items. This enhancement aims to make *FusionIndex* more comprehensive and inclusive, potentially leading to a more

well-rounded evaluation metric in future research endeavors. Finally, some prominent oversampling strategies used in traditional machine learning approaches can be used to determine the value of the ratings injected into tail items.

## ACKNOWLEDGEMENTS

In the preparation of certain sections of this manuscript, the authors utilized ChatGPT to improve grammar, enhance clarity, and support language refinement. All content generated with the assistance of this tool was subsequently reviewed, revised, and approved by the authors, who take full responsibility for the final version of the manuscript.

### Funding

This study is supported by the Scientific and Technical Research Council of Turkey (TUBITAK) under grant number 122E040. There was no additional external funding received for this study. The funders had no role in study design, data collection and analysis, decision to publish, or preparation of the manuscript.

### Grant Disclosures

The following grant information was disclosed by the authors:
Scientific and Technical Research Council of Turkey (TUBITAK): 122E040.

### Competing Interests

The authors declare that they have no competing interests.

### Author Contributions

- Mert Gulsoy conceived and designed the experiments, performed the experiments, performed the computation work, prepared figures and/or tables, authored or reviewed drafts of the article, and approved the final draft.
- Emre Yalcin conceived and designed the experiments, analyzed the data, performed the computation work, authored or reviewed drafts of the article, and approved the final draft.
- Alper Bilge conceived and designed the experiments, analyzed the data, performed the computation work, authored or reviewed drafts of the article, and approved the final draft.

### Data Availability

The source code and the datasets are available at GitHub and Zenodo:
- https://github.com/SiriusFoundation/EquiRate
- Yalcin, E. (2024). EquiRate: Balanced Rating Injection Approach for Popularity Bias Mitigation in Recommender Systems. https://doi.org/10.5281/zenodo.12515959.

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
