# Peer review of "EquiRate: balanced rating injection approach for popularity bias mitigation in recommender systems"

_PeerJ Computer Science, doi:10.7717/peerj-cs.3055_

## Round 0.1 · original submission · Major Revisions

The problem addressed by the paper is interesting, and the paper is clearly written. There are, however, limitations that should be addressed, mainly:
1) The related work section lacks references to relevant literature.
2) The experimental evaluations require clarifications and additional analysis.

Reviewer 1 ·

Basic reporting

The related work section should include a more thorough discussion of popularity debiasing in the context of Graph Neural Networks (GNNs), with references to studies such as [1].
It is recommended to provide additional details in Figure 2. Including a toy example that demonstrates the injection process would be beneficial for clarity.
[1] Zhou, Huachi, et al. "Adaptive popularity debiasing aggregator for graph collaborative filtering." Proceedings of the 46th International ACM SIGIR Conference on Research and Development in Information Retrieval. 2023.

Experimental design

1. No ablation study has been conducted to assess the contribution of each component.
2. Some experimental settings are unclear. For example, which recommendation models were used? Were they MLPs or GNNs?

Validity of the findings

No standard deviation or p-values are reported (e.g., in Tables 3 and 4), making it unclear whether the observed improvements are statistically significant.

Reviewer 2 ·

Basic reporting

- The manuscript is clearly written and employs professional, unambiguous English.
- The introduction provides sufficient background, establishing the context and importance of addressing popularity bias in recommender systems.
- References to related work are relevant and comprehensive, demonstrating awareness of existing solutions in the field.
- Figures are well-prepared, clear, and labeled adequately to support the narrative. The supplied raw data and methods align with the expectations for reproducibility.

Typos and Mistakes
1. "suffer from the popularity bias problem" should be "suffer from popularity bias" for conciseness.
2. Misaligned indentation in algorithms and equations, such as the layout in Algorithm 1, could improve clarity.
3. Equation formatting in the explanation of FusionIndex metrics appears inconsistent with inline text.

Suggestions for Improvement
- Provide additional justifications for parameter tuning (e.g., choice of α values in SRI).
- Clarify if the observed improvements hold across other recommendation contexts beyond the provided datasets.

Strengths
- Innovative and practical approach to mitigating popularity bias.
- Thorough evaluation using a range of metrics and real-world datasets.
- Clear demonstration of EquiRate’s impact on improving recommendation diversity.

Weaknesses
- Limited discussion of computational resource requirements.
- Insufficient exploration of the applicability of EquiRate to other recommendation scenarios.

Experimental design

- The research is original and fits the journal's scope. The problem of popularity bias in recommender systems is well-articulated, filling a gap in current methodologies.
- The experimental design is rigorous. Detailed descriptions of the datasets and parameters ensure the results are reproducible.
- Synthetic Rating Injection (SRI) and Generation (SRG) strategies are innovative, supported by theoretical underpinnings, and explained with clarity.
- The evaluation includes robust metrics, including the novel FusionIndex, which captures both accuracy and beyond-accuracy dimensions.

Validity of the findings

- The findings are consistent with the stated objectives and are statistically significant. Variants of the EquiRate method outperform benchmarks in both accuracy and diversity measures.
- The proposed FusionIndex metric effectively evaluates multidimensional performance, providing nuanced insights beyond single-criteria evaluation.
- Conclusions are well-supported, but additional discussion of limitations (e.g., computational complexity or generalizability to real-world systems) would enhance credibility.

Additional comments

The manuscript is a substantial contribution to addressing popularity bias in recommender systems. The proposed pre-processing approach (EquiRate) and the novel FusionIndex evaluation metric provide valuable tools for advancing fairness and diversity in recommendations. However, readability could benefit from improved formatting of technical sections, particularly algorithms and equations.

Reviewer 3 ·

Basic reporting

This paper introduces a novel data preprocessing module, EquiRate, to mitigate popularity bias in recommender systems. The method works by injecting artificial ratings for less popular or "tail" items. There are three closely related strategies (OPA, HIFA, and TPA) that are introduced to decide how ratings to inject for each item, and three strategies (GRV, PRV, RVG) to decide which rating to assign to fake interactions. Furthermore, the authors define a new performance metric, the FusionIndex, which incorporates both recommendation performance and various metrics of diversity: the FusionIndex is the arithmetic mean of four closely related indices, which are each the harmonic mean of nDCG and each diversity metric (APLT, Novelty, LTC, and Entropy). Experiments on ML1M, Yelp, and Douban books show that the method successfully improves the FusionIndex, with mixed improvements in the various diversity metrics.

The paper, especially the introduction, is very well written. The organization is clear (although a bit wordy) and professional. The explanation of the model is self-contained and reasonably clear. Although there are a couple of areas for improvement (as mentioned below), this is a decent submission and I am happy to recommend minor revisions.

Experimental design

The experimental design is generally clear, but there are a couple of possible issues that I would like the authors to answer.

(1) The authors haven't gone into a lot of detail describing the train-test split strategy they use. The standard approach would be to hold out a random set of interactions as the test set, and use the base model to rank all items except those in the training set, and compare with the relevance scores given by the test set. However, I do not understand how this can be reconciled with the definition of "Novelty" given by the authors. Indeed, it is claimed that "novelty" is the proportion of items recommended which are not already been interacted with, but in most reasonable train-test split strategies, these items would be specifically excluded from the ranking. Perhaps the metric is defined ignoring that constraint, but the train-test split strategy still excludes the training items?

(2) Similarly to above, there are many variants being evaluated, and even in terms of the FusionIndex, they are rarely all superior to all baselines. Some of these "variants" are just different hyperparameter values. The most reasonable way to evaluate against the baselines would be to incorporate a validation set to pick the hyperparameters and compare the best-performing method (chosen with the validation set) with the baselines on the test set.

(3, important) One thing that really confuses me in the main description of the method on page 8 is that one goes straight from the SRI strategy to the SRG strategy: this explains how to decide how many ratings to add to each item and how to pick the ratings themselves, but not which users to assign the ratings to. A more detailed mathematical explanation is needed here: are there additional interactions assigned to random users, or does another CF method rely on to decide which (user, item) pairs to add?

(4) The tables should incorporate boldface. I understand this will result in it being more obvious that the method doesn't outperform the other baselines in the core components, such as novelty, APLT, etc. However, this is not a huge problem as the method is still doing better in terms of FusionIndex. For instance, LNSF significantly outperforms the proposed method for all four diversity metrics, but doesn't do well in terms of recommendation performance, which is consistent with the tradeoff explained in the paper.

Validity of the findings

The conclusions are well-described and generally reasonable.

Additional comments

Medium:

(1) The work [8] is described as "closely related" to the present work. However, the difference isn't described in enough detail. To the best of my understanding, the method isn't a baseline either (although [9] is two of them, to the best of my understanding).

(2) Although I understand the authors have somewhat pertinently explained that postprocessing methods do not address the root cause of the bias in the way that preprocessing methods such as the present work do, it would still be would be recommended to incorporate postprocessing baselines from the fairness literature [4,5].

(3) Similarly, this is a preprocessing method that can be applied to any downstream RecSys algorithm; thus, one possible downside is the fact that only VAECF has been used. It would be good to compare the improvements offered by the method with other CF methods, such as fundamental matrix completion methods [2,7], methods which incorporate some user-side information [6], or self-supervised graph methods such as LightGCN.

(4) The definition of the FusionIndex is slightly unintuitive: why not simply take the (weighted) harmonic mean of NGCG, novelty, entropy, LPT, APLT? It can be weighted with weights 1/2, 1/6, 1/6, 1/6 to maintain the appropriate equal weightage on recommender performance and diversity metrics.

(5) How do you pick sigma and sigma max in RVG?


Minor:

(1) The algorithm (page 9) could be better structured. The way it is written, it appears that a separate SRG strategy could be used for each item.

(2) Formulae such as equations (7) and (9) are a little lacking in rigour. |i\in T\cap N_u| should be |T\cap N_u| or |\{i\in T\cap N_u \}. Similarly, the numerator in equation 9 could be written |N_u\cap I_u^c|.

(3) line 619 (page 15): the name of the section is missing

(4) line 614 (page 15): presents" should be "present

(5) line 658 (page 17), "Particularly, ..." could be replaced by "In particular, ..."

(6) line 792 (page 23), the section name is missing again

(7) line 804 (page 23), same problem with the missing section name.

(8) line 820 (page 23) "Yelp's dataset" could be "The Yelp dataset"

(9) line 415 (page 10): section name missing again

(10) Line 431, what is beta? It doesn't seem to be introduced anywhere else in the paper.

======

References:

[1] Liang et al. "Variational Autoencoders for Collaborative Filtering", WWW2018.

[2] Mazumder et al. "Spectral Regularization Algorithms for Learning Large Incomplete Matrices", JMLR 2010

[3] He et al. LightGCN: Simplifying and Powering Graph Convolution Network for Recommendation, SIGIR 2020

[4] Lopes et al. "Recommendations with minimum exposure guarantees: A post-processing framework", ESWA 2024

[5] "CPFair: Personalized Consumer and Producer Fairness Re-ranking for Recommender Systems", SIGIR 2022.

[6] Alves et al. "Uncertainty-adjusted recommendation via matrix factorization with weighted losses", TNNLS 2024

[7] Kasalicky et al. "Uncertainty-adjusted Inductive Matrix Completion with Graph Neural Networks", RecSys LBR 2023

[8] Yalcin and Bilge, "Evaluating unfairness of popularity bias in recommender systems: A comprehensive user-centric analysis", 2022, Information Process and Management.

[9] Yalcin and Bilge, "Investigating and counteracting popularity bias in group recommendations", Information Processing and Management. 2021

---

## Round 0.2 · accepted · Accept

The paper has been revised according to the reviewers recommendations and can be now accepted for publication.

**PeerJ Staff Note**: Although the Academic and Section Editors are happy to accept your article as being scientifically sound, a final check of the manuscript shows that it would benefit from further editing. Therefore, please identify necessary edits and address these while in proof stage. ( Minor errors in mathematical writing)

Reviewer 2 ·

Basic reporting

The authors addressed all the comments.

Experimental design

Now ok

Validity of the findings

Ok

Additional comments

Nop

Reviewer 3 ·

Basic reporting

As mentioned in my previous review, the paper is very well written and presents a large amount of reasonable work.

Experimental design

As mentioned previously, although the experimental design is generally sound, there were a couple of minor issues in the previous version, including:

A Lack of details for the train-test set procedure,

B Lack of boldface in the tables,

C The definition of "novelty".

D The comparison to the related works

E. Minor errors in mathematical writing


A,C: has been addressed mostly: the authors now clearly mention the user-wise cross-validation step. However, I don't understand what the authors mean in the rebuttal when they say "because the test user has no interactions in the training set, the recommender ranks all items in the catalog". How is this possible if the background method is VAECF: what is the input to the trained autoencoder, since there is no side information, this presumably must consist in some previous interactions, which would contradict this statement.


B: solved, thanks!

E: solved, thanks!

D. Mostly solved, thanks. I accept that the amount of effort is sufficient to make it prohibitively onerous to run variants of the model including other backbone methods. However, some justification of the choice of backbone (VAECF) and contextualisation would be nice. In such a long paper, there is definitely room.

Validity of the findings

The findings are generally valid (though not particularly impactful), and I appreciate the extra work that was done in the revision adding statistical significance and boldface in the tables. I recommend acceptance.

Additional comments

Regarding the lack of a validation set, thanks for the reply. In summary, you each individual method and hyperparameter configuration is evaluated with user-level LOACV. I understand that your focus isn't on actual performance but also on the tradeoff with novelty, etc., so it's perfectly fine and somewhat common. However, in other venues or in a context where performance matters more, some stricter reviewers would still interpret this as test-set leakage (if only one of your multiple methods outperforms the baseline it could still be due to random fluctuations). However, one can also argue that there are nearly as many baselines as methods of your own (which would introduce the possibility of the baselines outperforming you due to a random fluctuation between them), which makes the comparison fairer.

I am recommending "accept as is" because I don't think it makes sense to go through another reviewing round and the paper is in decent shape already. But if there is an option to make very minor changes for the CR version, you could incorporate a discussion of this as well as some of the points above (and in particular an answer to my question about the VAECF input).